# EXPLORING CHEMICAL SPACE WITH SCORE-BASED OUT-OF-DISTRIBUTION GENERATION

## ABSTRACT

A well-known limitation of existing molecular generative models is that the generated molecules highly resemble those in the training set. To generate truly novel molecules with completely different structures that may have even better properties than known molecules for *de novo* drug discovery, more powerful exploration in the chemical space is necessary. To this end, we propose *Molecular Out-Of-distribution Diffusion* (MOOD), a novel score-based diffusion scheme that incorporates out-of-distribution (OOD) control in the generative stochastic differential equation (SDE) with simple control of a hyperparameter, thus requires no additional computational costs unlike existing methods (e.g., RL-based methods). However, some novel molecules may be chemically implausible, or may not meet the basic requirements of real-world drugs. Thus, MOOD performs conditional generation by utilizing the gradients from a property prediction network that guides the reverse-time diffusion process to high-scoring regions according to multiple target properties such as protein-ligand interactions, drug-likeness, and synthesizability. This allows MOOD to search for novel and meaningful molecules rather than generating unseen yet trivial ones. We experimentally validate that MOOD is able to explore the chemical space beyond the training distribution, generating molecules that outscore ones found with existing methods, and even the top 0.01% of the original training pool.

## 1 INTRODUCTION

Finding novel molecules with desired chemical properties is the primary goal of drug discovery. However, the chemical space is vast, and it is infeasible to examine all possible molecules to find those satisfying a target molecule profile. Recently, deep molecule generation models that can automatically generate candidate molecules arose as promising substitutes (Gómez-Bombarelli et al., 2016; Lim et al., 2018; Schwalbe-Koda & Gómez-Bombarelli, 2019) for conventional experimental drug discovery approaches via trial-and-error processes with human efforts. However, most existing molecule generation models have the following two limitations, which limit their practical impact.

First of all, the common pitfall of the models based on distributional learning is that the exploration is confined to the training distribution, and the generated molecules highly resemble those in the training set. For example, Walters & Murcko (2020) point out that the top-scoring molecule found by the model of Zhavoronkov et al. (2019) exhibits "striking similarity" to known active molecules included in the training set (see Figure 1 (Left; a1, a2)). This highly limits its applicability to *de novo* drug discovery which aims to find completely new molecules rather than slight variations of existing ones, emphasizing the need for a generation strategy that can generate out-of-distribution (OOD) molecules with desired properties.

Secondly, there exists a discrepancy between the target chemical properties of the molecule generation models and those in real-world scenarios. The most common properties utilized by the molecule generation models are penalized logP and quantitative estimate of drug-likeness (QED) (Jin et al., 2018; You et al., 2018; Shi et al., 2019; Zang & Wang, 2020; Luo et al., 2021c; Liu et al., 2021). However, as criticized by Coley (2020), Cieplinski et al. (2020), and Xie et al. (2020), optimization of these scores may not lead to the discovery of useful drugs. For example, the top-scoring molecule found in terms of penalized logP in the state-of-the-art model is a trivial long chain of the maximum number of carbons (Luo et al., 2021c), since penalized logP prefers large molecules.

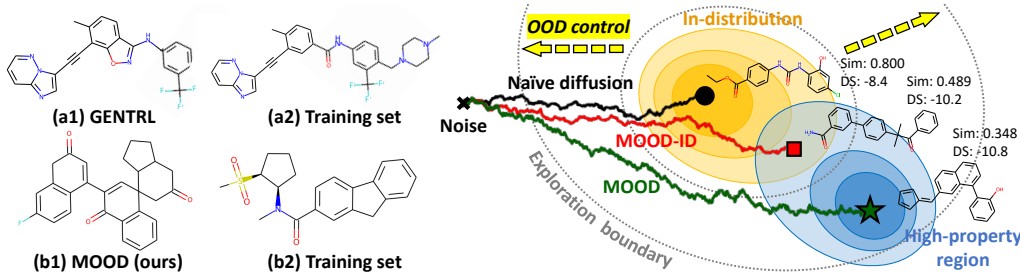

Figure 1: **(Left) The molecules found by GENTRL (Zhavoronkov et al., 2019) and MOOD, and the most similar molecules to those from the training set.** Unlike GENTRL, MOOD discovered a novel molecule that is different from any training molecule with a higher docking score than the top 0.01% of the training set. **(Right) Illustration of the reverse-time diffusion process of MOOD.** MOOD leverages the OOD-controlled diffusion to extend the exploration boundary and generate OOD samples in the low-density region, while using the property prediction network to guide the sampling process to the high-property region, thereby discovering molecules with desired target properties that lie beyond the training distribution. MOOD-ID is the variant of MOOD that only utilizes the property prediction network without the OOD control.

To overcome such a limitation of conventional property objectives, a few recent works adopted the *docking score*, a binding affinity score based on the three-dimensional simulation of a target protein and a drug candidate (Cieplinski et al., 2020). However, using the docking score as a sole metric is still insufficient as a reasonable proxy for drug activity, since heavy molecules with high docking scores are likely to be false positives due to the dependency of the docking score on molecular weights (Pan et al., 2003). Furthermore, real-world drug discovery involves searching for molecules that meet multiple requirements, for example, protein-ligand interactions, drug-likeness, and synthesizability.

Unfortunately, the poor explorability of most existing drug discovery methods makes it difficult to successfully accomplish the multi-objective tasks. As the number of chemical requirements increases, fewer molecules in the training set will satisfy the given constraints, and the optimization problem will become more difficult when trying to generate molecules that meet all the requirements. Thus, to generate high-scoring molecules with respect to multiple chemical properties, and further, that are applicable to the real-world, we need a method that can more effectively explore the chemical space.

To this end, we propose a novel *de novo* drug discovery framework for generating OOD molecules, that are completely different from those in the training set, but nonetheless satisfy the given constraints. Specifically, we first propose a score-based generative model for OOD generation, by deriving a novel OOD-controlled reverse-time diffusion process that can control the amount of deviation from the data distribution. However, since the naïve OOD generation can yield molecules that are chemically implausible, difficult to synthesize, and lacking desired properties, we further extend our framework to perform conditional generation for property optimization. Our *Molecular Out-Of-distribution Diffusion* (MOOD) framework utilizes the gradient of a property prediction network to guide the sampling process to domains that are highly likely to satisfy the given constraints, while leveraging the proposed OOD control to explore beyond the space of known molecules. MOOD is able to generate molecules that lie beyond the training distribution without additional computational costs, unlike existing methods (e.g., RL-based exploration methods).

We experimentally validate the proposed MOOD on the molecule optimization task, on which MOOD outperforms state-of-the-art molecule generation methods by generating novel molecules with high docking scores while satisfying QED and synthetic accessibility (SA) conditions, demonstrating its ability to effectively explore the chemical space and find chemical optima of multiple requirements. Notably, MOOD discovered a novel molecule (Figure 1 (Left; b1)) with a higher docking score than the top 0.01% of the training dataset. We summarize our contributions as follows:

- We devise a novel score-based generative model for OOD generation, which overcomes the limited explorability of previous generative models by leveraging our proposed OOD-controlled reverse-time diffusion process that can control the amount of deviation from the data distribution.

- We propose a novel score-based generative framework for molecule optimization which leverages the gradients of the property prediction network to guide the generation process, while extending the exploration space with the OOD control.

- We experimentally demonstrate that our proposed conditional OOD molecule generation framework can generate novel molecules that are drug-like, synthesizable, and have high docking scores on five protein targets, outperforming existing molecule generation methods, and even discovering novel molecules that outscore the top molecules in the original dataset.

## 2 RELATED WORK

**Score-based generative models**    Score-based generative models learn to reverse the perturbation process from data to noise in order to generate samples from a given data distribution. Song & Ermon (2019) proposed to design the perturbation process with multiple noise scales, while Ho et al. (2020) proposed to consider the perturbation as a parameterized Markov chain, and Song et al. (2021b) united these approaches by generalizing the perturbation as a diffusion process modeled by a stochastic differential equation (SDE). Recently, score-based generative models have shown successful results for the generation of graphs (Niu et al., 2020; Jo et al., 2022), and arose as promising methods for molecular conformation generation (Shi et al., 2021; Xu et al., 2022; Luo et al., 2021b) and even 3D molecule generation (Hoogeboom et al., 2022). Yet, the application of score-based models for targeted *de novo* drug discovery poses a unique challenge not found in other domains: finding novel molecules that satisfy specific constraints in the vast chemical space. To the best of our knowledge, we are the first to propose a score-based generative framework for molecule optimization.

**Conditional score-based models**    Recently, score-based generative models have been applied to conditional generation tasks, such as image inpainting (Song et al., 2021b), super-resolution (Choi et al., 2021; Li et al., 2022; Saharia et al., 2021), MRI reconstruction (Chung & Ye, 2021; Jalal et al., 2021; Song et al., 2021a), image translation (Meng et al., 2021; Sasaki et al., 2021), and point cloud generation (Lyu et al., 2021). However, directly adapting these schemes to molecule optimization is challenging, due to the complex dependency between nodes and edges which determines the chemical validity and properties of molecules. We introduce a novel conditional reverse-time diffusion for controlled OOD generation, while using a property predictor to guide the sampling process, which together steers the generation to the intersection of low-density and high-property regions.

**Molecule generation models**    Existing methods for generating molecular graphs of desired properties include models based on variational autoencoders (VAEs) (Gómez-Bombarelli et al., 2018; Jin et al., 2018; Liu et al., 2018; Eckmann et al., 2022), generative adversarial networks (GANs) (Lima Guimaraes et al., 2017; De Cao & Kipf, 2018), genetic algorithms (Jensen, 2019), and flow-based models (Shi et al., 2019; Zang & Wang, 2020; Luo et al., 2021c). A score-based graph generation model for unconditional molecule generation has been proposed recently (Jo et al., 2022). A common shortcoming of existing works that are based on distributional learning or fragment vocabularies is the limited exploration in the chemical space beyond the known data distribution, as they focus on interpolating the learned distribution or reassembling substructures of known molecules. Among the few works that consider docking score as the target property (Olivecrona et al., 2017; Jeon & Kim, 2020; Yang et al., 2021; Eckmann et al., 2022), Yang et al. (2021) focus on the exploration and proposed an exploration-promoting RL objective to discover novel molecules. However, exploration of the agent is computationally expensive, and the method is inherently limited by the fragment vocabulary, which are the subgraphs of the seen molecules. Contrarily, our framework is able to generate novel molecules with desired properties outside the distribution of the training set, without requiring high computational costs or a fragment vocabulary.

## 3 MOLECULE OPTIMIZATION WITH SCORE-BASED OUT-OF-DISTRIBUTION GENERATION

In this section, we introduce our Molecular Out-Of-distribution Diffusion (MOOD) framework, which aims to generate molecules that are both novel with respect to the training data distribution and have desired chemical properties. We first present a novel OOD-controlled diffusion process that can explore beyond the training distribution in Section 3.1. Then, we describe our proposed MOOD based on a property-guided sampling process with OOD-controlled diffusion in Section 3.2. We begin with the descriptions of molecular graphs and the score-based graph generation scheme.

### 3.1 SCORE-BASED OUT-OF-DISTRIBUTION GENERATION

**Molecular graph representation**    A molecule can be represented as a molecular graph $G = (\boldsymbol{X}, \boldsymbol{A}) \in \mathbb{R}^{N \times F} \times \mathbb{R}^{N \times N} \coloneqq \mathcal{G}$, where $\boldsymbol{X}$ is the node feature matrix carrying the information of the atom types described by $F$-dimensional one-hot encoding, $\boldsymbol{A}$ is the adjacency matrix representing the bond types, and $N$ denotes the maximum number of heavy atoms (i.e., atoms besides hydrogen) of a molecule in the dataset. This representation directly uses the bond types (1 for single bonds, 2 for double bonds, and 3 for triple bonds) as elements of $\boldsymbol{A}$ instead of the one-hot encoding.

**Score-based graph generation** The seminal work of Song et al. (2021b) models the diffusion from data to noise through a stochastic differential equation (SDE), and learns to reverse the process from noise to data. However, its naïve extension to graph generation cannot model the complex dependency between nodes and edges, which is crucial for learning the distribution of graphs. To address this problem, Jo et al. (2022) proposed Graph Diffusion via the System of SDEs (GDSS), which models the diffusion of both the node features and the adjacency matrix with a system of SDEs. Specifically, the forward diffusion for a graph $\{\boldsymbol{G}_t = (\boldsymbol{X}_t, \boldsymbol{A}_t)\}_{t=0}^{T}$ is defined by an Itô SDE:

$$\mathrm{d}\boldsymbol{G}_t = \mathbf{f}_t(\boldsymbol{G}_t)\mathrm{d}t + g_t\mathrm{d}\mathbf{w}, \tag{1}$$

with the linear drift coefficient $\mathbf{f}_t(\cdot)\colon \mathcal{G} \to \mathcal{G}^1$, the scalar diffusion coefficient $g_t\colon \mathcal{G} \to \mathbb{R}$, and the standard Wiener process $\mathbf{w}$. Denoting the marginal distribution under the forward diffusion as $p_t$, the corresponding reverse diffusion process can be described by the following system of SDEs:

$$\begin{cases} \mathrm{d}\boldsymbol{X}_t = \left[\mathbf{f}_{1,t}(\boldsymbol{X}_t) - g_{1,t}^2\nabla_{\boldsymbol{X}_t}\log p_t(\boldsymbol{X}_t, \boldsymbol{A}_t)\right]\mathrm{d}\bar{t} + g_{1,t}\mathrm{d}\bar{\mathbf{w}}_1 \\ \mathrm{d}\boldsymbol{A}_t = \left[\mathbf{f}_{2,t}(\boldsymbol{A}_t) - g_{2,t}^2\nabla_{\boldsymbol{A}_t}\log p_t(\boldsymbol{X}_t, \boldsymbol{A}_t)\right]\mathrm{d}\bar{t} + g_{2,t}\mathrm{d}\bar{\mathbf{w}}_2, \end{cases} \tag{2}$$

where $\mathbf{f}_t(\boldsymbol{X}, \boldsymbol{A}) = (\mathbf{f}_{1,t}(\boldsymbol{X}), \mathbf{f}_{2,t}(\boldsymbol{A}))$ and $g_t = (g_{1,t}, g_{2,t})$ are the drift and diffusion coefficients, respectively, $\bar{\mathbf{w}}_1$ and $\bar{\mathbf{w}}_2$ are the reverse-time standard Wiener processes, and $\mathrm{d}\bar{t}$ is an infinitesimal negative time step. The score networks $\boldsymbol{s}_{\theta_1,t}$ and $\boldsymbol{s}_{\theta_2,t}$ are trained to approximate the partial score functions $\nabla_{\boldsymbol{X}_t}\log p_t(\boldsymbol{X}_t, \boldsymbol{A}_t)$ and $\nabla_{\boldsymbol{A}_t}\log p_t(\boldsymbol{X}_t, \boldsymbol{A}_t)$, respectively, then used to simulate Eq. (2) backwards in time to jointly generate the node features and the adjacency matrices.

Although GDSS can generate high-quality molecular graphs that follow the data distribution, it is not free from the explorative limitation of deep generative models described in Section 1. To tackle this limitation, we introduce a novel score-based OOD generative model.

**Exploration with OOD control** To expand the exploration space of the diffusion, we propose a novel OOD-controlled score-based graph generative model that can generate samples outside in-distribution, where the OOD-ness of the generative process is controlled by the hyperparameter $\lambda \in [0, 1)$. We approach by sampling from the conditional distribution $p_t(\boldsymbol{G}_t|\mathbf{y}_o = \lambda)$ where $\mathbf{y}_o$ represents the OOD condition, by solving the following conditional reverse-time SDE:

$$\mathrm{d}\boldsymbol{G}_t = \left[\mathbf{f}_t(\boldsymbol{G}_t) - g_t^2\nabla_{\boldsymbol{G}_t}\log p_t(\boldsymbol{G}_t|\mathbf{y}_o = \lambda)\right]\mathrm{d}\bar{t} + g_t\mathrm{d}\bar{\mathbf{w}}. \tag{3}$$

The conditional score $\nabla_{\boldsymbol{G}_t}\log p_t(\boldsymbol{G}_t|\mathbf{y}_o = \lambda)$ can be decomposed as the sum of the two gradients:

$$\nabla_{\boldsymbol{G}_t}\log p_t(\boldsymbol{G}_t|\mathbf{y}_o = \lambda) = \nabla_{\boldsymbol{G}_t}\log p_t(\boldsymbol{G}_t) + \nabla_{\boldsymbol{G}_t}\log p_t(\mathbf{y}_o = \lambda|\boldsymbol{G}_t), \tag{4}$$

and since the score function $\nabla_{\boldsymbol{G}_t}\log p_t(\boldsymbol{G}_t)$ can be estimated by the score networks $\boldsymbol{s}_{\theta_1,t}$ and $\boldsymbol{s}_{\theta_2,t}$, simulating Eq. (3) is possible if the second term is known. In order to access $\nabla_{\boldsymbol{G}_t}\log p_t(\mathbf{y}_o = \lambda|\boldsymbol{G}_t)$, we exploit the fact that the OOD samples are the ones of low-likelihood with respect to the in-distribution (Du & Mordatch, 2019; Grathwohl et al., 2020). Specifically, we propose to model the distribution $p_t(\mathbf{y}_o = \lambda|\boldsymbol{G}_t)$ to be proportional to the negative exponent of the density $p_t(\boldsymbol{G}_t)^2$:

$$p_t(\mathbf{y}_o = \lambda|\boldsymbol{G}_t) \propto p_t(\boldsymbol{G}_t)^{-\sqrt{\lambda}} \cdot \mathbb{1}_{\boldsymbol{H}_t}, \tag{5}$$

where $\boldsymbol{H}_t = \{\boldsymbol{G}_t \,|\, p_t(\boldsymbol{G}_t) \geq p_*\}$ and $p_*$ is the probability threshold. The existence of such distribution is shown in Section A.1 of the appendix.

Based on the modeling of Eq. (5), we derive a novel OOD-controlled reverse-time diffusion process from Eq. (3) as follows (see Section A.2 of the appendix for the detailed derivation):

$$\mathrm{d}\boldsymbol{G}_t = \left[\mathbf{f}_t(\boldsymbol{G}_t) - (1 - \sqrt{\lambda})g_t^2\nabla_{\boldsymbol{G}_t}\log p_t(\boldsymbol{G}_t)\right]\mathrm{d}\bar{t} + g_t\mathrm{d}\bar{\mathbf{w}}. \tag{6}$$

Intuitively, as $\lambda$ approaches 1, the distribution $p_t(\mathbf{y}_o = \lambda|\boldsymbol{G}_t)$ modeled by Eq. (5) becomes sharper since the negative exponent induces larger magnitude for smaller probability values, which amplifies the effect of the OOD condition. Accordingly, the influence of the score $\nabla_{\boldsymbol{G}_t}\log p_t(\boldsymbol{G}_t)$ in the drift coefficient weakens, and the sampling process is guided to the lower density regions.

---

[1]$t$-subscript is used to represent a function of time: $F_t(\cdot) := F(\cdot, t)$ and $\boldsymbol{M}_{\theta,t}(\cdot) := \boldsymbol{M}_\theta(\cdot, t)$.
[2]We empirically found that using $\sqrt{\lambda}$ instead of $\lambda$ yields well-scaled results as the value of $\lambda$ changes.

Looking from the perspective of the reverse-time diffusion process, Eq. (6) induces a marginal distribution proportional to $p_t(\boldsymbol{G}_t)^{1-\sqrt{\lambda}}$. Consequently, simulating the OOD-controlled diffusion process backward in time generates samples from the relatively uniform distribution compared to the original data distribution, where the dispersion is controlled by $\lambda$, and the corresponding samples are more likely to come from outside the in-distribution. Therefore, the proposed OOD-controlled diffusion process of Eq. (6) can be used as an OOD generative model that can control the deviation from the data distribution with the hyperparameter $\lambda$. Notably, the OOD control enables us to explore further from the data distribution without additional computational costs, in contrast to previous molecule generation methods (Olivecrona et al., 2017; Jeon & Kim, 2020; Yang et al., 2021) that rely on costly reinforcement learning algorithms for exploration.

We empirically demonstrate that the proposed OOD-controlled diffusion process is indeed able to control the OOD-ness of the generated samples on a simple Gaussian mixture in Figure 2. While the OOD control with $\lambda = 0$ (i.e., GDSS) accurately generates samples from the data distribution, we can generate a wide scope of OOD samples by simply increasing the hyperparameter $\lambda$.

However, being able to generate OOD molecules does not necessarily mean that we will be able to discover useful molecules, since they may be chemically implausible, difficult to synthesize, or have low affinity to a target protein. Thus, for the OOD generator to be truly useful, it should conditionally generate molecules that satisfy certain desired conditions, which we describe in Section 3.2.

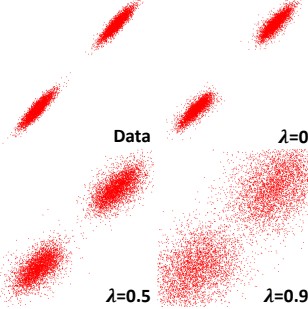

Figure 2: **A toy experiment on the OOD-controlled diffusion.**

## 3.2 MOLECULE PROPERTY OPTIMIZATION

**Property optimization with conditional generation** Our goal is to generate novel molecules that possess desired chemical properties, for example, high binding affinity against a target protein. If we represent the condition of maximizing certain property as $\mathbf{y}_p$, our objective then is to generate molecules from the conditional distribution $p_t(\boldsymbol{G}_t|\mathbf{y}_o = \lambda, \mathbf{y}_p)$, which can be decomposed as follows:

$$p_t(\boldsymbol{G}_t|\mathbf{y}_o = \lambda, \mathbf{y}_p) \propto p_t(\boldsymbol{G}_t)\, p_t(\mathbf{y}_o = \lambda|\boldsymbol{G}_t)\, p_t(\mathbf{y}_p|\boldsymbol{G}_t, \mathbf{y}_o = \lambda). \tag{7}$$

Since $p_t(\mathbf{y}_p|\boldsymbol{G}_t, \mathbf{y}_o = \lambda)$ represents the probability that the molecular graph $\boldsymbol{G}_t$ satisfies the property $\mathbf{y}_p$, we propose to model the probability density using the Boltzmann distribution as follows:

$$p_t(\mathbf{y}_p|\boldsymbol{G}_t, \mathbf{y}_o = \lambda) = e^{\alpha_t P_\phi(\boldsymbol{G}_t, \lambda)}/Z_t, \tag{8}$$

where $\alpha_t$ is the scaling coefficient, $Z_t$ is the normalization constant, and $P_\phi$ is the property function estimated by a property prediction network, which we describe in detail at the end of this section.

Using Eq. (5) and Eq. (8), we propose a novel conditional reverse-time diffusion process for generating OOD molecules that satisfy specific constraints as follows:

$$d\boldsymbol{G}_t = \left[\mathbf{f}_t(\boldsymbol{G}_t) - (1 - \sqrt{\lambda})g_t^2 \nabla_{\boldsymbol{G}_t} \log p_t(\boldsymbol{G}_t) - \alpha_t g_t^2 \nabla_{\boldsymbol{G}_t} P_\phi(\boldsymbol{G}_t, \lambda)\right] d\bar{t} + g_t d\bar{\mathbf{w}}, \tag{9}$$

which we refer to as *Molecular Out-Of-distribution Diffusion* (MOOD). Figure 1 (Right) illustrates the generation process of MOOD, where the additional gradients $\sqrt{\lambda} g_t^2 \nabla_{\boldsymbol{G}_t} \log p_t(\boldsymbol{G}_t)$ and $-\alpha_t g_t^2 \nabla_{\boldsymbol{G}_t} P_\phi(\boldsymbol{G}_t, \lambda)$ of Eq. (9), that are not in the unconditional process of Eq. (2), can be understood as the guidance that drive the sampling process to the low-density regions and the high-property regions, respectively. However, Eq. (9) cannot be directly used as a generative model since it does not model the node-edge relationships (Jo et al., 2022), and thus we utilize the equivalent diffusion process through the system of reverse-time SDEs as follows:

$$\begin{cases} d\boldsymbol{X}_t = \left[\mathbf{f}_{1,t}(\boldsymbol{X}_t) - (1 - \sqrt{\lambda})g_{1,t}^2\, \boldsymbol{s}_{\theta_1,t}(\boldsymbol{X}_t, \boldsymbol{A}_t) - \alpha_{1,t} g_{1,t}^2 \nabla_{\boldsymbol{X}_t} P_\phi(\boldsymbol{X}_t, \boldsymbol{A}_t, \lambda)\right] d\bar{t} + g_{1,t} d\bar{\mathbf{w}}_1 \\ d\boldsymbol{A}_t = \left[\mathbf{f}_{2,t}(\boldsymbol{A}_t) - (1 - \sqrt{\lambda})g_{2,t}^2\, \boldsymbol{s}_{\theta_2,t}(\boldsymbol{X}_t, \boldsymbol{A}_t) - \alpha_{2,t} g_{2,t}^2 \nabla_{\boldsymbol{A}_t} P_\phi(\boldsymbol{X}_t, \boldsymbol{A}_t, \lambda)\right] d\bar{t} + g_{2,t} d\bar{\mathbf{w}}_2 \,. \end{cases} \tag{10}$$

To balance the effect of the OOD control and the property gradient, we propose to automatically set $\alpha_{1,t}$ and $\alpha_{2,t}$ throughout the diffusion according to a predefined ratio between the magnitudes of the partial scores and the property gradients as follows:

$$\alpha_{1,t} = r_{1,t} \frac{\|\boldsymbol{s}_{\theta_1,t}(\boldsymbol{G}_t)\|}{\|\nabla_{\boldsymbol{X}_t} P_\phi(\boldsymbol{G}_t, \lambda)\|} \quad , \quad \alpha_{2,t} = r_{2,t} \frac{\|\boldsymbol{s}_{\theta_2,t}(\boldsymbol{G}_t)\|}{\|\nabla_{\boldsymbol{A}_t} P_\phi(\boldsymbol{G}_t, \lambda)\|}, \tag{11}$$

where $r_{1,t}$ and $r_{2,t}$ are the time-dependent magnitude ratios and $\|\cdot\|$ is the entry-wise matrix norm.

Figure 3: **(Left) UMAP visualization** of the ZINC250k dataset and the generated molecules by the proposed OOD-controlled diffusion process. Each point represents a molecule based on the activation of the ChemNet layer. **(Right) Evaluation results of the molecules generated by the OOD-controlled diffusion.** We report FCD, NSPDK MMD, and novelty of the generated molecules with various values of the hyperparameter $\lambda$.

**Property prediction network**   To approximate the property function of the desired property, we train a property prediction network $P_\phi$ to estimate the property of a given molecule $\boldsymbol{G}_t$. Since chemical properties are entirely determined by molecules, $P_\phi$ can predict the target property without $\lambda$, and we utilize the architecture of the discriminator network of De Cao & Kipf (2018) as follows:

$$P_\phi(\boldsymbol{G}_t) \coloneqq \text{MLP}_s(\tanh(\boldsymbol{H}')) \ \text{ for } \ \boldsymbol{H}' = \text{MLP}_s\left(\left[\{\boldsymbol{H}_j\}_{j=0}^L\right]\right) \odot \text{MLP}_t\left(\left[\{\boldsymbol{H}_j\}_{j=0}^L\right]\right), \quad (12)$$

where $\boldsymbol{H}_{i+1} = \tanh(\text{GNN}(\boldsymbol{H}_i, \boldsymbol{A}_t))$ with a graph convolutional network (GCN) (Kipf & Welling, 2017) as the GNN and $\boldsymbol{H}_0 = \boldsymbol{X}_t$, $L$ is the number of the GNN operations, $\text{MLP}_s$ and $\text{MLP}_t$ are the multilayer perceptrons (MLPs) with sigmoid and tanh activation functions, respectively, $\odot$ is the element-wise multiplication, and $[\cdot]$ is the concatenation operation.

## 4  EXPERIMENTS

We first validate the efficacy of our OOD-controlled diffusion process on a novel molecule generation task in Section 4.1, then demonstrate the effectiveness of MOOD on property optimization tasks in Section 4.2. We further conduct an ablation study to verify the effectiveness of MOOD's individual components in Section 4.3.

### 4.1  NOVEL MOLECULE GENERATION

**Experimental setup**   To verify that our proposed OOD-controlled diffusion scheme can control the OOD-ness of the generated samples and enhance the explorability, we first conduct an experiment on an unconstrained novel molecule generation task. We generate 3,000 molecules without incorporating the property network (i.e., Eq. (6)), varying the hyperparameter $\lambda$ for OOD control. We measure the OOD-ness of the generated molecules with respect to the training dataset, ZINC250k (Irwin et al., 2012), using the following metrics. **Fréchet ChemNet Distance (FCD)** (Preuer et al., 2018) is the distance between the training and generated set of molecules based on the activations from the penultimate layer of a ChemNet. **Neighborhood subgraph pairwise distance kernel maximum mean discrepancy (NSPDK MMD)** (Costa & De Grave, 2010) is the MMD between the test set and the generated set. **Novelty** (Jin et al., 2020b; Xie et al., 2020) is the fraction of valid molecules that have a similarity less than 0.4 with the nearest neighbor in the training set.

**Results**   We visualize the distribution of the generated molecules via two-dimensional uniform manifold approximation and projection (UMAP) (McInnes & Healy, 2018) in Figure 3 (Left). We observe that the proposed OOD-controlled diffusion scheme not only enables to generate OOD molecules, but also allows the deviation from the training dataset to be controllable with the hyperparameter $\lambda$. As the value of $\lambda$ increases, the sampling space deviates more from the training distribution. We further quantitatively measure the OOD-ness of the generated molecules in Figure 3 (Right). Similarly, as $\lambda$ increases, FCD and NSPDK MMD increase, which shows that the distribution of generated molecules becomes more different from the training distribution in the view of biochemical activity and molecular graph structures. Notably, larger $\lambda$ increases novelty as well, indicating that each generated molecule is indeed comprised of chemically distinct substructures from the seen molecules.

### 4.2  PROPERTY OPTIMIZATION

**Experimental setup**   The goal of the property optimization task is to generate novel molecules that are of high binding affinity, drug-like, and synthesizable. To reflect these constraints, we accordingly construct the property function $P_{obj}$ as follows:

$$P_{obj}(\boldsymbol{G}_t) = \widehat{\text{DS}}(\boldsymbol{G}_t) \times \text{QED}(\boldsymbol{G}_t) \times \widehat{\text{SA}}(\boldsymbol{G}_t) \in [0, 1], \quad (13)$$

Table 1: **Novel hit ratio (%) results.** The results are the means and the standard deviations of 5 runs. The best performance and comparable results ($p > 0.05$) are highlighted in bold.

| Method | Target protein | | | | |
|---|---|---|---|---|---|
| | parp1 | fa7 | 5ht1b | braf | jak2 |
| REINVENT (Olivecrona et al., 2017) | 0.480 $(\pm 0.344)$ | 0.213 $(\pm 0.081)$ | 2.453 $(\pm 0.561)$ | 0.127 $(\pm 0.088)$ | 0.613 $(\pm 0.167)$ |
| JTVAE (Jin et al., 2018) | 0.856 $(\pm 0.211)$ | 0.289 $(\pm 0.016)$ | 4.656 $(\pm 1.406)$ | 0.144 $(\pm 0.068)$ | 0.815 $(\pm 0.044)$ |
| GraphAF (Shi et al., 2019) | 0.689 $(\pm 0.166)$ | 0.011 $(\pm 0.016)$ | 3.178 $(\pm 0.393)$ | 0.956 $(\pm 0.319)$ | 0.767 $(\pm 0.098)$ |
| HierVAE (Jin et al., 2020a) | 0.553 $(\pm 0.214)$ | 0.007 $(\pm 0.013)$ | 0.507 $(\pm 0.278)$ | 0.207 $(\pm 0.220)$ | 0.227 $(\pm 0.127)$ |
| GraphDF (Luo et al., 2021c) | 0.044 $(\pm 0.031)$ | 0.000 $(\pm 0.000)$ | 0.000 $(\pm 0.000)$ | 0.011 $(\pm 0.016)$ | 0.011 $(\pm 0.016)$ |
| FREED (Yang et al., 2021) | 3.627 $(\pm 0.961)$ | **1.107** $(\pm 0.209)$ | 10.187 $(\pm 3.306)$ | 2.067 $(\pm 0.626)$ | 4.520 $(\pm 0.673)$ |
| FREED-QS | 4.627 $(\pm 0.727)$ | **1.332** $(\pm 0.113)$ | 16.767 $(\pm 0.897)$ | 2.940 $(\pm 0.359)$ | 5.800 $(\pm 0.295)$ |
| LIMO (Eckmann et al., 2022) | 0.455 $(\pm 0.057)$ | 0.044 $(\pm 0.016)$ | 1.189 $(\pm 0.181)$ | 0.278 $(\pm 0.134)$ | 0.689 $(\pm 0.319)$ |
| GDSS (Jo et al., 2022) | 1.933 $(\pm 0.208)$ | 0.368 $(\pm 0.103)$ | 4.667 $(\pm 0.306)$ | 0.167 $(\pm 0.134)$ | 1.167 $(\pm 0.281)$ |
| MOOD-w/o $P_\phi$ (ours) | 2.127 $(\pm 0.216)$ | 0.447 $(\pm 0.091)$ | 7.900 $(\pm 0.455)$ | 0.520 $(\pm 0.117)$ | 2.293 $(\pm 0.223)$ |
| MOOD-ID (ours) | 3.400 $(\pm 0.117)$ | 0.433 $(\pm 0.063)$ | 11.873 $(\pm 0.521)$ | 2.207 $(\pm 0.165)$ | 3.953 $(\pm 0.383)$ |
| MOOD (ours) | **7.017** $(\pm 0.428)$ | 0.733 $(\pm 0.141)$ | **18.673** $(\pm 0.423)$ | **5.240** $(\pm 0.285)$ | **9.200** $(\pm 0.524)$ |

Table 2: **Novel top 5% docking score (kcal/mol) results.** The results are the means and the standard deviations of 5 runs. The best performance and comparable results ($p > 0.05$) are highlighted in bold.

| Method | Target protein | | | | |
|---|---|---|---|---|---|
| | parp1 | fa7 | 5ht1b | braf | jak2 |
| REINVENT (Olivecrona et al., 2017) | -8.702 $(\pm 0.523)$ | -7.205 $(\pm 0.264)$ | -8.770 $(\pm 0.316)$ | -8.392 $(\pm 0.400)$ | -8.165 $(\pm 0.277)$ |
| JTVAE (Jin et al., 2018) | -9.482 $(\pm 0.132)$ | -7.683 $(\pm 0.048)$ | -9.382 $(\pm 0.332)$ | -9.079 $(\pm 0.069)$ | -8.885 $(\pm 0.026)$ |
| GraphAF (Shi et al., 2019) | -9.327 $(\pm 0.030)$ | -7.084 $(\pm 0.025)$ | -9.113 $(\pm 0.126)$ | -9.896 $(\pm 0.226)$ | -8.267 $(\pm 0.101)$ |
| HierVAE (Jin et al., 2020a) | -9.487 $(\pm 0.278)$ | -6.812 $(\pm 0.274)$ | -8.081 $(\pm 0.252)$ | -8.978 $(\pm 0.525)$ | -8.285 $(\pm 0.370)$ |
| GraphDF (Luo et al., 2021c) | -6.823 $(\pm 0.134)$ | -6.072 $(\pm 0.081)$ | -7.090 $(\pm 0.100)$ | -6.852 $(\pm 0.318)$ | -6.759 $(\pm 0.111)$ |
| FREED (Yang et al., 2021) | -10.427 $(\pm 0.177)$ | **-8.297** $(\pm 0.094)$ | -10.425 $(\pm 0.331)$ | -10.325 $(\pm 0.164)$ | -9.624 $(\pm 0.102)$ |
| FREED-QS | -10.579 $(\pm 0.104)$ | **-8.378** $(\pm 0.044)$ | -10.714 $(\pm 0.183)$ | -10.561 $(\pm 0.080)$ | -9.735 $(\pm 0.022)$ |
| LIMO (Eckmann et al., 2022) | -8.984 $(\pm 0.223)$ | -6.764 $(\pm 0.142)$ | -8.422 $(\pm 0.063)$ | -9.046 $(\pm 0.316)$ | -8.435 $(\pm 0.273)$ |
| GDSS (Jo et al., 2022) | -9.967 $(\pm 0.028)$ | -7.775 $(\pm 0.039)$ | -9.459 $(\pm 0.101)$ | -9.224 $(\pm 0.068)$ | -8.926 $(\pm 0.089)$ |
| MOOD-w/o $P_\phi$ (ours) | -10.086 $(\pm 0.038)$ | -7.932 $(\pm 0.054)$ | -9.838 $(\pm 0.083)$ | -9.634 $(\pm 0.052)$ | -9.247 $(\pm 0.041)$ |
| MOOD-ID (ours) | -10.409 $(\pm 0.030)$ | -7.947 $(\pm 0.034)$ | -10.487 $(\pm 0.069)$ | -10.421 $(\pm 0.050)$ | -9.575 $(\pm 0.075)$ |
| MOOD (ours) | **-10.865** $(\pm 0.113)$ | -8.160 $(\pm 0.071)$ | **-11.145** $(\pm 0.042)$ | **-11.063** $(\pm 0.034)$ | **-10.147** $(\pm 0.060)$ |

where $\widehat{DS}$ is the normalized docking score (DS), QED is the drug-likeness, and $\widehat{SA}$ is the normalized synthetic accessibility (SA). We train the property network $P_\phi$ to predict the property value $P_{obj}$ of the molecules in the ZINC250k dataset. Previously used metrics for evaluating docking-optimized molecules, such as hit ratio or the average of the top 5% DS (Yang et al., 2021) are insufficient for *de novo* drug discovery, as they do not consider whether the generated molecules are novel or not. Therefore, we evaluate 3,000 generated molecules with the following metrics. **Novel hit ratio (%)** is the fraction of unique hit molecules that have the maximum Tanimoto similarity less than 0.4 with the training molecules. Here, *hit molecules* are defined as the molecules that satisfy the following constraints: DS < (the median DS of the known active molecules), QED > 0.5, and SA < 5. **Novel top 5% docking score** is the average DS of the top 5% unique molecules that satisfy the constraints QED > 0.5 and SA < 5 and have the maximum similarity less than 0.4 with the training molecules. Note that these metrics jointly evaluate novelty and multiple properties, thereby more suitable for real-world scenarios. We use five protein targets, **parp1**, **fa7**, **5ht1b**, **braf**, and **jak2**.

**Baselines** **REINVENT** (Olivecrona et al., 2017) is an RL model that utilizes a prior sequence model. **JTVAE** (Jin et al., 2018) is a VAE-based model that utilizes the junction tree molecular representation and Bayesian optimization. **GraphAF** (Shi et al., 2019) and **GraphDF** (Luo et al., 2021c) are flow-based models that utilize continuous and discrete latent variables, respectively. **HierVAE** (Jin et al., 2020a) is a VAE-based model that utilizes hierarchical molecular representation and active learning. **FREED** (Yang et al., 2021) is a fragment-based RL model that utilizes prioritized experience replay (PER) (Schaul et al., 2015) for exploration, and **FREED-QS** is our modification of FREED that exploits $P_{obj}$ of Eq. (13) as its reward function. **LIMO** (Eckmann et al., 2022) is a VAE-based model that generates molecules with an inceptionism-like technique. We also compare with **GDSS** (Jo et al., 2022), **MOOD-w/o $P_\phi$**, MOOD that only utilizes the OOD exploration without the property prediction network by setting $r_{1,t} = r_{2,t} = 0$, and **MOOD-ID**, MOOD that only utilizes the property prediction network without the OOD exploration by setting $\lambda = 0$.

**Results** As shown in Table 1 and Table 2, our proposed MOOD significantly outperforms all the baselines for most of the target proteins, and shows competitive results on fa7 compared to FREED and FREED-QS. The performance gap shown in the novel hit ratio and the novel top 5% DS indicates that MOOD is superior in discovering novel molecules that are drug-like, synthesizable, and have a high binding affinity. Notably, MOOD consistently outperforms MOOD-ID for all the target proteins,

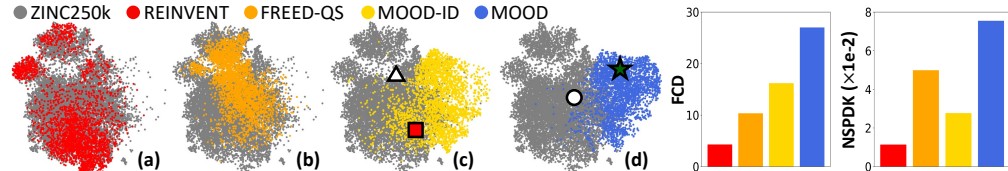

Figure 4: **(Left) UMAP visualization of the molecules** from ZINC250k and the generated samples with parp1 as the target protein. See Figure 5 for the symbols depicted in (c) and (d). **(Right) Distributional distances of the generated molecules** measured by FCD and NSPDK MMD with respect to ZINC250k.

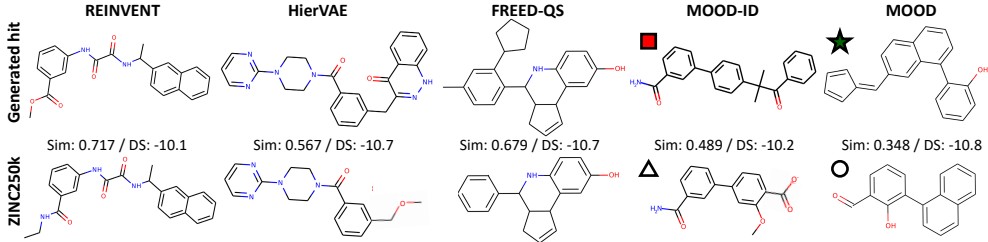

Figure 5: **Generated hit molecules with parp1 as the target protein and the corresponding ZINC250k molecules of the highest similarity.** The similarity and docking score (kcal/mol) are provided at the bottom of each generated hit. The molecules with symbols are the ones marked in Figure 4 (c) and (d).

and MOOD-w/o $P_\phi$ also consistently outperforms GDSS even without the aid of property gradient, together demonstrating that the proposed exploration via OOD generation is highly effective in finding novel chemical optima of multiple constraints. We further provide the results of novelty, diversity, uniqueness, hit ratio, and top 5% DS in Table 9, Table 10, Table 11, Table 12, and Table 13 of Section D.2, respectively. As shown in these tables, the OOD control utilized in MOOD largely enhances novelty while maintaining near-perfect uniqueness compared to MOOD-ID. Moreover, the improved exploration over MOOD-ID also aids MOOD in terms of hit ratio and top 5% DS, as it is able to discover molecules with better properties outside the data distribution.

**Explorability** We visualize the distribution of the generated molecules via UMAP in Figure 4 (Left). As shown in the figure, MOOD exhibits superior explorability beyond the training distribution compared to the baselines. While the generated molecules of REINVENT and FREED-QS lie close to the ZINC250k molecules, most of the generated molecules of MOOD lie beyond the training distribution. Note that MOOD-ID generates some molecules that deviate from the training distribution by the effect of the property gradient in its diffusion, yet the fraction is smaller than those of MOOD due to the lack of the OOD constraint. We further measure the distributional distance of the generated molecules from ZINC250k via FCD and NSPDK MMD in Figure 4 (Right). Together with the high novelty shown in Table 9 of Section D.2, the results verify that MOOD is able to generate novel molecules that are significantly different from those of the training dataset.

**Generated molecules** We visualize the generated molecules of MOOD and the baselines in Figure 5 and Figure 8 of Section D.3. As shown in the figures and also in Table 9 of Section D.2, the molecules generated by the baselines possess duplicated substructures with the ZINC250k molecules due to the limited explorability and accordingly have high maximum Tanimoto similarity and low novelty. This limits their application to *de novo* drug discovery. In contrast, the generated molecules of MOOD exhibit low similarity with the ZINC250k molecules, while having high binding affinity. As shown in Figure 4, the molecule found by MOOD in Figure 5 is indeed an OOD sample, that lies outside the training distribution, unlike the one found by MOOD-ID.

**Discovery with MOOD** To validate that MOOD can find novel chemical optima that lie beyond the training distribution with even better chemical properties, we visualize the hit molecules found by MOOD that have higher binding affinity to parp1 than those of the top 0.01% of ZINC250k in Figure 6. Note that the molecules also have low similarity with the molecules of ZINC250k. This result suggests the applicability and efficacy of MOOD in real-world *de novo* drug discovery.

Figure 6: **Novel hit molecules found by MOOD** against parp1 and the top 0.01% ZINC250k molecules.

Table 3: **Novel hit ratio (%) and novel top 5% docking score (kcal/mol)** results with 3D molecule generation baselines and GDSS with respect to the target protein glmu. Results are the means and the standard deviations of 5 runs. Best performance and its comparable results ($p > 0.05$) are highlighted in bold.

| Method | Novel hit ratio | Novel top 5% DS |
|---|---|---|
| Luo et al. (2021a) | 1.367 (± 1.324) | -6.328 (± 0.567) |
| Pocket2Mol (Peng et al., 2022) | 6.002 (± 0.913) | -7.714 (± 0.123) |
| GDSS (Jo et al., 2022) | 1.227 (± 0.100) | -6.411 (± 0.046) |
| MOOD-w/o $P_\phi$ (ours) | 7.320 (± 0.404) | -7.673 (± 0.069) |
| MOOD-ID (ours) | 10.453 (± 1.811) | -7.832 (± 0.169) |
| MOOD (ours) | **16.733** (± 1.984) | **-8.423** (± 0.164) |

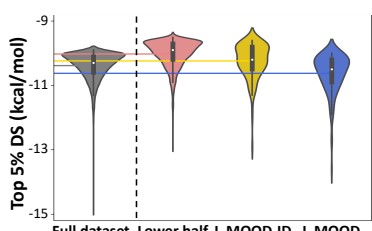

Figure 7: **Top 5% docking score distribution of the molecules** with respect to the target protein parp1. Each of the horizontal lines represents the average value of the distribution.

**Comparison with 3D molecule generation methods**    We additionally compare MOOD with the methods of a recently emerging area, namely, three-dimensional (3D) molecule generation. The model of Luo et al. (2021a) and Pocket2Mol (Peng et al., 2022) are autoregressive models that generate molecules of high binding affinity by utilizing 3D information of the binding site of the target protein. As shown in Table 3, MOOD largely outperforms the baselines even without the spatial information of the binding pocket, again confirming that MOOD is highly practical and has great potential in solving real-world drug discovery problems.

### 4.3    ABLATION STUDY

**Effects of the OOD control and property gradient**    To examine the effect of the proposed OOD control and property gradient, we compare MOOD-w/o $P_\phi$, MOOD-ID and MOOD with GDSS. As shown in Table 1, Table 2, and Table 3, using both the OOD generation scheme and the guidance from the property prediction network is essential for finding better chemical optima. Specifically, the superior generation result of MOOD over MOOD-w/o $P_\phi$ and MOOD-ID over GDSS demonstrate the effectiveness of the property gradient, while the superiority of MOOD over MOOD-ID and MOOD-w/o $P_\phi$ over GDSS demonstrate the effectiveness of the OOD control.

**Training on the low-property subset**    To further validate the explorability of the proposed MOOD, we evaluate the generated molecules of L-MOOD-ID and L-MOOD, which are respectively MOOD-ID and MOOD trained on the lower half subset of the ZINC250k dataset in terms of $P_{obj}$ (Eq. (13)). We visualize the distribution of the top 5% DS of the molecules that satisfy QED > 0.5 and SA < 5 in Figure 7. We first observe that both L-MOOD-ID and L-MOOD yield higher top 5% DS compared to their training set, demonstrating the effectiveness of the proposed conditional diffusion process with the property prediction network. Furthermore, unlike L-MOOD-ID, L-MOOD is able to generate molecules with higher top 5% DS compared to the original ZINC250k dataset, even though L-MOOD has never seen the molecules in the higher half of ZINC250k. This shows that exploring beyond the known chemical space with MOOD is not only beneficial for novel molecule discovery, but also for property optimization since the novel molecules may better satisfy the given properties.

## 5    CONCLUSION

To tackle the limited explorability of previous molecule generation models, we proposed *Molecular Out-Of-distribution Diffusion* (MOOD), a new score-based generative model for generating novel molecules with desired chemical properties outside the training distribution. MOOD leverages a novel OOD-controlled reverse-time diffusion process that can control the OOD-ness of the generated samples without any additional costs. MOOD further incorporates a novel conditional score-based diffusion scheme to optimize molecules for target chemical properties, using the gradient of the property prediction network to guide the diffusion process to the high-property regions. We validated MOOD on the multi-objective property optimization tasks that are analogous to real-world drug discovery scenarios, on which ours outperforms existing molecular generation methods with superior explorability and property optimization performance. Further, our method generated novel molecules that are largely different from any of the existing molecules in the training set, showing its potential as a promising means of *de novo* drug discovery.

## REPRODUCIBILITY STATEMENT

- **Code.** We provide the codebase to construct MOOD and conduct the experiments in the paper as the supplementary file.
- **Datasets.** We use ZINC250k (Irwin et al., 2012) and CrossDocked2020 (Francoeur et al., 2020) for the experiments.
- **Toy experiment.** We provide the details of the toy experiment of Figure 2 in Section C.1.
- **Novel molecule generation.** We provide the details of the novel molecule generation experiment of Figure 3 in Section C.3.
- **Property optimization.** We provide the details of the property optimization experiment of Table 1, Table 2, Figure 4, Figure 5, Figure 6, and Figure 7 in Section C.4. The details of the property optimization experiment of Table 3 with the CrossDocked2020 dataset are also provided in Section C.4.

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

# A    PROOFS AND DERIVATIONS

## A.1    EXISTENCE OF THE OOD-CONTROLLED DISTRIBUTION

It is enough to show that there exists a probability distribution $q$ satisfying the following:

$$q(\boldsymbol{G}) \propto p(\boldsymbol{G})^{-\sqrt{\lambda}} \cdot \mathbb{1}_{\boldsymbol{H}}, \tag{14}$$

where $\boldsymbol{H} = \{\boldsymbol{G} \,|\, p(\boldsymbol{G}) \geq p_*\}$ and $\lambda \in [0, 1)$. Let $\tilde{q} = p(\boldsymbol{G})^{-\sqrt{\lambda}} \cdot \mathbb{1}_{\boldsymbol{H}}$. Using the fact that $p(\cdot)$ is a probability distribution defined in $\mathcal{G}$,

$$Z = \int_{\mathcal{G}} \tilde{q}(\boldsymbol{G}) \mathrm{d}\boldsymbol{G} = \int_{\boldsymbol{H}} p(\boldsymbol{G})^{-\sqrt{\lambda}} \mathrm{d}\boldsymbol{G} \leq \int_{\boldsymbol{H}} p_*^{-\sqrt{\lambda}-1} p(\boldsymbol{G}) \mathrm{d}\boldsymbol{G} \leq p_*^{-\sqrt{\lambda}-1} < \infty. \tag{15}$$

Therefore, defining the probability $q$ as $q = \tilde{q}/Z$ shows the existence of such distribution.

## A.2    DERIVING THE OOD-CONTROLLED DIFFUSION PROCESS

Using Eq. (5) to model $p_t(\mathbf{y}_o = \lambda | \boldsymbol{G}_t)$ to be proportional to the negative exponent of the density $p_t(\boldsymbol{G}_t)$, we can derive the following reverse-time SDE from Eq. (3):

$$\mathrm{d}\boldsymbol{G}_t = \left[\mathbf{f}_t(\boldsymbol{G}_t) - (1 - \sqrt{\lambda}) g_t^2 \nabla_{\boldsymbol{G}_t} \log p_t(\boldsymbol{G}_t) - g_t^2 \nabla_{\boldsymbol{G}_t} \log \mathbb{1}_{\boldsymbol{H}_t}\right] \mathrm{d}\bar{t} + g_t \mathrm{d}\bar{\mathbf{w}}. \tag{16}$$

For sufficiently small threshold $p_*$, we obtain $\nabla_{\boldsymbol{G}_t} \log \mathbb{1}_{\boldsymbol{H}_t} = 0$, thus the generation process simulated by the reverse-time SDE in Eq. (16) is equivalent to the generation process simulated by the following reverse-time SDE:

$$\mathrm{d}\boldsymbol{G}_t = \left[\mathbf{f}_t(\boldsymbol{G}_t) - (1 - \sqrt{\lambda}) g_t^2 \nabla_{\boldsymbol{G}_t} \log p_t(\boldsymbol{G}_t)\right] \mathrm{d}\bar{t} + g_t \mathrm{d}\bar{\mathbf{w}}, \tag{17}$$

which corresponds to the proposed OOD-controlled diffusion process in Eq. (6) of the main paper.

# B    ADDITIONAL REMARK ON RELATED WORK

Note that recently, Sehwag et al. (2022) introduced a conditional score-based model for generation of images from low-density regions, by modifying the sampling process using a discriminative model to steer the generative process to the low-density region. However, it is clearly different from our proposed OOD control scheme described in Section 3.1, as it 1) requires an additionally trained model to sample from the low-density region, and moreover, 2) cannot control the amount of deviation of the generated samples from the in-distribution.

# C    EXPERIMENTAL DETAILS

## C.1    TOY EXPERIMENT

Following Jo et al. (2022), we utilize a bivariate Gaussian mixture as the data distribution of the toy experiment in Section 3.1 as follows:

$$p_{data}(\mathbf{x}) = \mathcal{N}(\mathbf{x} \,|\, \mu_1, \Sigma_1) + \mathcal{N}(\mathbf{x} \,|\, \mu_2, \Sigma_2), \tag{18}$$

where the mean and variance are given as follows:

$$\mu_1 = \begin{pmatrix} 0.5 \\ 0.5 \end{pmatrix}, \ \mu_2 = \begin{pmatrix} -0.5 \\ -0.5 \end{pmatrix},$$

$$\Sigma_1 = \Sigma_2 = 0.1^2 \begin{pmatrix} 1.0 & 0.9 \\ 0.9 & 1.0 \end{pmatrix}.$$

We set the number of linear layers as 20 with residual paths, the hidden dimension as 512, the type of SDEs as VPSDE with $\beta_{min} = 0.01$ and $\beta_{max} = 0.05$, the number of training epochs as 5000, the batch size as 2048, and use an Adam optimizer (Kingma & Ba, 2014). We generate $2^{13}$ samples in Figure 2 with a PC sampler of a signal-to-noise ratio (SNR) of 0.05 and a scale coefficient of 0.8.

## C.2 UMAP VISUALIZATION

To produce the UMAP visualization of the generated molecules in Figure 4, we first randomly select 5,000 molecules from the ZINC250k dataset and 3,000 molecules from REINVENT, FREED-QS, MOOD-ID, and MOOD which are generated against the target protein parp1, respectively. ChemNet (Preuer et al., 2018) activations are computed from the molecules and then together visualized by the UMAP library (McInnes & Healy, 2018).

## C.3 NOVEL MOLECULE GENERATION

**Measuring the novelty**   We measure the novelty of the generated molecules as the fraction of valid molecules with similarity less than 0.4 compared to the nearest neighbor $G_{\text{SNN}}$ in the training dataset, which can be formally written as follows:

$$\frac{1}{n} \sum_{G \in \mathcal{M}} \mathbb{1}\{\text{sim}(G, G_{\text{SNN}}) < 0.4\}, \tag{19}$$

where $\mathcal{M}$ is the set of $n$ valid molecules, and $\text{sim}(G, G')$ is the pairwise Tanimoto similarity over Morgan fingerprints of radius 2 and 1024 bits.

**Implementation details**   Following Jo et al. (2022), each molecule is preprocessed into a graph of $X \in \{0, 1\}^{N \times F}$ and $A \in \{0, 1, 2, 3\}^{N \times N}$, where $N$ is the maximum number of atoms in a molecule of the dataset, and $F$ is the number of possible atom types. The elements of $A$ indicate the bond types (single, double, or triple). All molecules are preprocessed to their kekulized form and all hydrogens are removed by the RDKit (Landrum et al., 2016) library. We utilize the valency correction proposed by Zang & Wang (2020). We use the pretrained score networks $s_{\theta_1}$ and $s_{\theta_2}$ from Jo et al. (2022)[3], which are trained on the ZINC250k (Irwin et al., 2012) dataset with the same train/test split used by Kusner et al. (2017). Following the original paper, we use VP and VE SDEs for diffusion of node and adjacency matrices, respectively, and set the SNR and the scale coefficient as 0.2 and 0.8, respectively. As in Jo et al. (2022), we quantize the entries of the adjacency matrices by mapping the values of $(-\infty, 0.5)$ to 0, the values of $[0.5, 1.5)$ to 1, the values of $[1.5, 2.5)$ to 2, and the values of $[2.5, +\infty)$ to 3 after the sampling.

## C.4 PROPERTY OPTIMIZATION

**Scoring function**   We use the popular docking program QuickVina 2 (Alhossary et al., 2015) to compute the docking scores and set the exhaustiveness as 1, following Yang et al. (2021). Note that the docking scores are negative values. QED and SA scores are computed using the RDKit (Landrum et al., 2016) library. To compute the objective function of Eq. (13), we clip the docking score in the range $[-20, 0]$ and compute $\widehat{\text{DS}}$ and $\widehat{\text{SA}}$ as follows:

$$\widehat{\text{DS}} = -\frac{\text{DS}}{20}, \quad \widehat{\text{SA}} = \frac{10 - \text{SA}}{9}. \tag{20}$$

Following Yang et al. (2021), we choose five proteins, **parp1** (Poly [ADP-ribose] polymerase-1), **fa7** (Coagulation factor VII), **5ht1b** (5-hydroxytryptamine receptor 1B), **braf** (Serine/threonine-protein kinase B-raf), and **jak2** (Tyrosine-protein kinase JAK2), that have highest AUROC scores when the protein-ligand binding affinities for DUD-E ligands are approximated with AutoDock Vina, as the target proteins about which the docking scores are calculated.

**Scheduling the scaling coefficients**   Instead of manually setting the value of the scaling coefficients of the Boltzmann distribution $\alpha_{1,t}$ and $\alpha_{2,t}$, we automatically set the value through the time-dependent magnitude ratios $r_{1,t}$ and $r_{2,t}$, respectively, throughout the diffusion process as shown in Eq. (11). The ratios are in turn scheduled according to time as follows:

$$r_{1,t} = r_{1,0} \cdot 0.1^t, \quad r_{2,t} = r_{2,0} \cdot 0.1^t, \tag{21}$$

where $r_{1,0}$ and $r_{2,0}$ are the final values (when $t = 0$) of $r_{1,t}$ and $r_{2,t}$, respectively.

---

[3]https://github.com/harryjo97/GDSS

**Evaluation metrics**   Novel hit ratio is the fraction of unique hit molecules that have the maximum Tanimoto similarity less than 0.4 with the training molecules, which can be written as follows:

$$\frac{1}{n} \sum_{\boldsymbol{G} \in \mathcal{M}} \mathbb{1}\{\text{DS}(\boldsymbol{G}) > (\text{the median DS of the known active molecules}), \text{QED}(\boldsymbol{G}) > 0.5, \text{SA}(\boldsymbol{G}) < 5\}, \tag{22}$$

where $n$ is the number of total generated molecules, and $\mathcal{M}$ is the set of generated molecules with no duplicates. Novel top 5% docking score is the average DS of the top 5% of total generated molecules with no duplicates that satisfy the following constraints: QED > 0.5, SA < 5, and $\text{sim}(\boldsymbol{G}, \boldsymbol{G}_{\text{SNN}}) < 0.4$. $\boldsymbol{G}_{\text{SNN}}$ is the nearest training molecule of the generated molecule $\boldsymbol{G}$, and $\text{sim}(\boldsymbol{G}, \boldsymbol{G}')$ is the pairwise Tanimoto similarity over Morgan fingerprints of radius 2 and 1024 bits.

**Implementation details**   The implementation details regarding the preprocessing of data, the model of GDSS, and the postprocessing procedure are the same as explained in Section C.3. We perform the grid search with the search space $\lambda \in \{0.01, 0.02, 0.03, 0.04, 0.05\}$, and find $\lambda = 0.04$ performs reasonably well throughout the experiments. Regarding the hyperparameters of the property prediction network, we set the number of the GNN operations $L$ as 3 with the hidden dimension of 16. The number of linear layers in $\text{MLP}_s$ and $\text{MLP}_t$ are both 1, and the number of linear layers in the final $\text{MLP}_s$ is 2. We perform the grid search with the search space $r_{1,0} \in \{0.3, 0.4, 0.5, 0.6, 0.7, 0.8\}$, and set $r_{1,0}$ as 0.5, 0.4, 0.6, 0.7, and 0.6 for the optimization with the target protein as parp1, fa7, 5ht1b, braf, and jak2, respectively. We find that $r_{2,0} = 0$ (i.e., $\alpha_{2,t} = 0$) performs reasonably well regardless of the target.

**Implementation details of the baselines**   We follow the corresponding original papers for most of the settings of the baselines. We describe the specifics and the differences from the original papers here. For REINVENT, we utilize the ZINC250k dataset to construct the vocabulary and train the prior[4]. For MORLD, we set the absolute value of docking score from QuickVina 2 as the final reward function and use benzene as the initial molecule[5]. For HierVAE, we use the ZINC250k dataset to construct the vocabulary and pretrain the model[6]. We follow the procedure utilized in Yang et al. (2021) to finetune the model for the optimization task. Specifically, we finetune the model using the DUD-E active molecules of the target protein as the training set, and set the number of training epochs as 800 for 5ht1b and 700 for other proteins, since the 5ht1b training set is larger. We adopt the two-cycle active learning scheme that gathers the generated hit molecules in the first round, then utilize them as additional training molecules in the second round. Note that although the active learning scheme improves the performance of HierVAE, it also requires twice of docking computations as the other methods. For FREED and FREED-QS, we utilize the predictive error-PER[7]. For LIMO, we utilized the pretrained VAE model and train the property network to predict the docking score from QuickVina 2, and generate molecules without the filtering based on the ring size for a fair evaluation[8].

**Implementation details of Table 3**   For the comparison with the 3D molecule generation baselines, we train GDSS and MOOD on the CrossDocked2020 (Francoeur et al., 2020) dataset with the same train/test split used in Luo et al. (2021a) and Peng et al. (2022). We utilize the molecules with less than 40 atoms as the training set, which corresponds to 96.77% of the original training set. We choose **glmu** (N-acetylglucosamine-1-phosphate uridyltransferase) as the target protein, and set the hit threshold as the docking score of the reference ligand contained in the test set. For the model of Luo et al. (2021a), we utilize the pretrained model without the duplication or ring filtering for a fair evaluation[9]. For Pocket2Mol, we utilize the pretrained model and generate molecules with 5,000 initial molecules, and collect the first 3,000 generated molecules without the duplication filtering for a fair evaluation[10].

---

[4]https://github.com/MarcusOlivecrona/REINVENT

[5]https://github.com/wsjeon92/morld

[6]https://github.com/wengong-jin/hgraph2graph

[7]https://github.com/AITRICS/FREED

[8]https://github.com/Rose-STL-Lab/LIMO

[9]https://github.com/luost26/3D-Generative-SBDD

[10]https://github.com/pengxingang/Pocket2Mol

Table 4: **Novel molecule generation results on the QM9 dataset.**

| $\lambda$ | FCD | NSPDK MMD ($\times 10^{-2}$) | Novelty (%) |
|---|---|---|---|
| 0 | 3.794 | 0.354 | 44.033 |
| 0.15 | 6.276 | 2.119 | 70.533 |

### C.5 COMPUTING RESOURCES

We conduct all the experiments on TITAN RTX, GeForce RTX 2080 Ti, or GeForce RTX 3090 GPUs.

## D ADDITIONAL EXPERIMENTAL RESULTS

### D.1 NOVEL MOLECULE GENERATION

To verify that the proposed OOD-controlled diffusion scheme can control the OOD-ness of the generated samples, we additionally conduct the novel molecule generation task on the QM9 (Ramakrishnan et al., 2014) dataset. We report the FCD, NSPDK MMD, and novelty results in Table 4.

### D.2 PROPERTY OPTIMIZATION

We report the property optimization results of additional baselines that are not based on distribution learning or a fragment vocabulary in Table 5 and Table 6. **GCPN** (You et al., 2018) is an atom-based RL model that utilizes adversarial training. **Graph GA** (Jensen, 2019) is a genetic algorithm-based model that utilizes predefined crossovers and mutations. **MORLD** (Jeon & Kim, 2020) is an RL model that uses QED and SA scores as intermediate rewards and docking scores as final rewards with the MolDQN algorithm (Zhou et al., 2019). As shown in the tables, MOOD outperforms the baselines in most target proteins, and this performance gap increases with the harsher novelty condition of 0.3 as shown in Table 7 and Table 8. These results show that MOOD is indeed very effective in generating drug candidates that are both novel and high-quality and superior in *de novo* drug discovery tasks even to the methods that do not conduct distribution learning or utilize a fragment vocabulary.

We also additionally report the novelty, diversity, uniqueness, hit ratio, and the top 5% docking score of the generated molecules in Table 9, Table 10, Table 11, Table 12, and Table 13. **Novelty** is the fraction of valid molecules with similarity less than 0.4 compared to the nearest neighbor in the training set, as explained in Section C.3. **Diversity** is calculated based on the pairwise similarity over Morgan fingerprints of the generated molecules. **Uniqueness** is the fraction of the valid molecules that are unique. **Hit ratio (%)** is the fraction of unique hit molecules. **Top 5% docking score** is the average DS of the top 5% unique molecules that satisfy the constraints QED > 0.5 and SA < 5. Note that the high novelty of MORLD shown in Table 9 arose from the fact that MORLD does not utilize any training molecules nor fragment vocabulary extracted from known molecules, and LIMO also exhibits high novelty due to its inceptionism-like technique applied to the learned latent space. However, the novelty values are trivial as MORLD and LIMO completely fail to generate high-quality and meaningful molecules. As one can observe in Table 1, Table 2, Table 12, and Table 13, the majority of the generated molecules of MORLD and LIMO do not satisfy the basic QED and SA constraints or do not exhibit high binding affinity. Also note that in the case of HierVAE, the validity is very low and since FCD and NSPDK MMD only take valid molecules into account, the FCD and NSPDK MMD values do not contain much information.

Table 5: **Novel hit ratio (%) results.** The results are the means and the standard deviations of 5 runs. The best performance and comparable results ($p > 0.05$) are highlighted in bold.

| Method | Target protein | | | | |
|---|---|---|---|---|---|
| | parp1 | fa7 | 5ht1b | braf | jak2 |
| GCPN (You et al., 2018) | 0.056 (± 0.016) | 0.444 (± 0.333) | 0.444 (± 0.150) | 0.033 (± 0.027) | 0.256 (± 0.087) |
| Graph GA (Jensen, 2019) | 4.811 (± 1.661) | 0.422 (± 0.193) | 7.011 (± 2.732) | **3.767** (± 1.498) | 5.311 (± 1.667) |
| MORLD (Jeon & Kim, 2020) | 0.047 (± 0.050) | 0.007 (± 0.013) | 0.880 (± 0.735) | 0.047 (± 0.040) | 0.227 (± 0.118) |
| MOOD (ours) | **7.017** (± 0.479) | **0.733** (± 0.141) | **18.673** (± 0.423) | **5.240** (± 0.285) | **9.200** (± 0.524) |

Table 6: **Novel top 5% docking score (kcal/mol) results.** The results are the means and the standard deviations of 5 runs. The best performance and comparable results ($p > 0.05$) are highlighted in bold.

| Method | Target protein | | | | |
|---|---|---|---|---|---|
| | parp1 | fa7 | 5ht1b | braf | jak2 |
| GCPN (You et al., 2018) | -7.464 (± 0.089) | -7.024 (± 0.629) | -7.632 (± 0.058) | -7.691 (± 0.197) | -7.533 (± 0.140) |
| Graph GA (Jensen, 2019) | **-10.949** (± 0.532) | -7.365 (± 0.326) | -10.422 (± 0.670) | **-10.789** (± 0.341) | **-10.167** (± 0.576) |
| MORLD (Jeon & Kim, 2020) | -7.532 (± 0.260) | -6.263 (± 0.165) | -7.869 (± 0.650) | -8.040 (± 0.337) | -7.816 (± 0.133) |
| MOOD (ours) | **-10.865** (± 0.113) | **-8.160** (± 0.071) | **-11.145** (± 0.042) | **-11.063** (± 0.034) | **-10.147** (± 0.060) |

Table 7: **Novel hit ratio (%) results** with the similarity condition of 0.3. The results are the means and the standard deviations of 5 runs. The best performance and comparable results ($p > 0.05$) are highlighted in bold.

| Method | Target protein | | | | |
|---|---|---|---|---|---|
| | parp1 | fa7 | 5ht1b | braf | jak2 |
| GCPN (You et al., 2018) | 0.044 (± 0.016) | **0.378** (± 0.333) | 0.344 (± 0.126) | 0.000 (± 0.000) | 0.222 (± 0.113) |
| Graph GA (Jensen, 2019) | 0.311 (± 0.134) | 0.011 (± 0.016) | 0.644 (± 0.340) | 0.267 (± 0.309) | 0.778 (±0.468) |
| MORLD (Jeon & Kim, 2020) | 0.040 (± 0.039) | 0.007 (± 0.013) | 0.760 (± 0.587) | 0.033 (± 0.037) | 0.207 (± 0.106) |
| MOOD (ours) | **4.107** (± 0.405) | **0.387** (± 0.163) | **10.687** (± 0.411) | **3.293** (± 0.351) | **5.525** (± 0.578) |

Table 8: **Novel top 5% docking score (kcal/mol) results** with the similarity condition of 0.3. The results are the means and the standard deviations of 5 runs. The best performance and comparable results ($p > 0.05$) are highlighted in bold.

| Method | Target protein | | | | |
|---|---|---|---|---|---|
| | parp1 | fa7 | 5ht1b | braf | jak2 |
| GCPN (You et al., 2018) | -7.347 (± 0.099) | -6.870 (± 0.579) | -7.445 (± 0.039) | -7.589 (± 0.210) | -7.426 (± 0.141) |
| Graph GA (Jensen, 2019) | -7.558 (± 0.173) | -5.423 (± 0.164) | -7.465 (± 0.558) | -8.059 (± 0.488) | -7.780 (± 0.465) |
| MORLD (Jeon & Kim, 2020) | -7.253 (± 0.198) | -6.037 (± 0.135) | -7.734 (± 0.570) | -7.572 (± 0.212) | -7.560 (± 0.187) |
| MOOD (ours) | **-10.585** (± 0.124) | **-7.740** (± 0.070) | **-10.817** (± 0.089) | **-10.754** (± 0.059) | **-9.876** (± 0.105) |

Table 9: **Novelty (%) results.** The results are the means and the standard deviations of 5 runs. The best results are highlighted in bold. The results of GDSS and MOOD-w/o $P_\phi$ are the same for the different target proteins.

| Method | Target protein | | | | |
|---|---|---|---|---|---|
| | parp1 | fa7 | 5ht1b | braf | jak2 |
| REINVENT (Olivecrona et al., 2017) | 9.894 (± 2.178) | 10.731 (± 1.516) | 11.605 (± 3.688) | 8.715 (± 2.712) | 11.456 (± 1.793) |
| MORLD (Jeon & Kim, 2020) | 98.433 (± 1.189) | 97.967 (± 1.764) | **98.787** (± 0.743) | 96.993 (± 2.787) | 97.720 (± 0.995) |
| HierVAE (Jin et al., 2020a) | 60.453 (±17.165) | 24.853 (±15.416) | 48.107 (± 1.988) | 59.747 (±16.403) | 85.200 (±14.262) |
| FREED (Yang et al., 2021) | 71.483 (± 1.233) | 57.687 (± 8.808) | 64.460 (±12.037) | 65.560 (±11.701) | 72.607 (± 5.170) |
| FREED-QS | 74.640 (± 2.953) | 78.787 (± 2.132) | 75.027 (± 5.194) | 73.653 (± 4.312) | 75.907 (± 5.916) |
| LIMO (Eckmann et al., 2022) | **99.356** (± 0.247) | **98.589** (± 0.042) | 94.267 (± 1.688) | **98.756** (± 0.220) | **98.911** (± 0.185) |
| GDSS (Jo et al., 2022) | 75.933 (± 0.427) | - | - | - | - |
| MOOD-w/o $P_\phi$ (ours) | 79.460 (± 0.221) | - | - | - | - |
| MOOD-ID (ours) | 72.607 (± 3.184) | 75.793 (± 1.377) | 70.321 (± 1.529) | 70.667 (± 1.024) | 69.947 (± 1.323) |
| MOOD (ours) | 84.180 (± 2.123) | 83.180 (± 1.519) | 84.613 (± 0.822) | 87.413 (± 0.830) | 83.273 (± 1.455) |

Table 10: **Diversity results.** The results are the means and the standard deviations of 5 runs. The best results are highlighted in bold. The results of GDSS and MOOD-w/o $P_\phi$ are the same for the different target proteins.

| Method | Target protein | | | | |
|---|---|---|---|---|---|
| | parp1 | fa7 | 5ht1b | braf | jak2 |
| REINVENT (Olivecrona et al., 2017) | 0.827 (± 0.007) | 0.842 (± 0.006) | 0.841 (± 0.006) | 0.831 (± 0.005) | 0.851 (± 0.004) |
| MORLD (Jeon & Kim, 2020) | **0.895** (± 0.001) | 0.893 (± 0.000) | **0.896** (± 0.001) | **0.893** (± 0.002) | **0.895** (± 0.001) |
| HierVAE (Jin et al., 2020a) | 0.724 (± 0.003) | 0.725 (± 0.002) | 0.739 (± 0.008) | 0.749 (± 0.003) | 0.762 (± 0.012) |
| FREED (Yang et al., 2021) | 0.831 (± 0.010) | 0.842 (± 0.009) | 0.831 (± 0.010) | 0.837 (± 0.008) | 0.808 (± 0.010) |
| FREED-QS | 0.855 (± 0.003) | 0.855 (± 0.002) | 0.850 (± 0.002) | 0.851 (± 0.003) | 0.850 (± 0.003) |
| LIMO (Eckmann et al., 2022) | 0.894 (± 0.002) | **0.898** (± 0.001) | 0.891 (± 0.002) | **0.893** (± 0.001) | 0.894 (± 0.001) |
| GDSS (Jo et al., 2022) | 0.887 (± 0.004) | - | - | - | - |
| MOOD-w/o $P_\phi$ (ours) | 0.886 (± 0.005) | - | - | - | - |
| MOOD-ID (ours) | 0.884 (± 0.003) | 0.887 (± 0.000) | 0.880 (± 0.002) | 0.875 (± 0.002) | 0.878 (± 0.001) |
| MOOD (ours) | 0.873 (± 0.005) | 0.889 (± 0.003) | 0.872 (± 0.003) | 0.862 (± 0.000) | 0.866 (± 0.001) |

Table 11: **Uniqueness (%) results.** The results are the means and the standard deviations of 5 runs. The best results are highlighted in bold. The results of GDSS and MOOD-w/o $P_\phi$ are the same for the different target proteins.

| Method | Target protein | | | | |
|---|---|---|---|---|---|
| | parp1 | fa7 | 5ht1b | braf | jak2 |
| REINVENT (Olivecrona et al., 2017) | 99.781 (± 0.265) | 99.780 (± 0.121) | 99.706 (± 0.161) | 99.663 (± 0.280) | 99.714 (± 0.407) |
| MORLD (Jeon & Kim, 2020) | 99.427 (± 0.666) | 99.320 (± 0.874) | 99.880 (± 0.086) | 99.367 (± 0.924) | 99.667 (± 0.173) |
| HierVAE (Jin et al., 2020a) | 4.480 (± 0.645) | 6.667 (± 0.967) | 4.707 (± 1.022) | 5.773 (± 0.931) | 4.053 (± 0.866) |
| FREED (Yang et al., 2021) | 97.153 (± 2.886) | 97.593 (± 1.877) | 95.133 (± 3.385) | 96.760 (± 2.601) | 96.667 (± 2.382) |
| FREED-QS | **100.000** (± 0.000) | **99.980** (± 0.040) | **100.000** (± 0.000) | **99.913** (± 0.173) | **99.940** (± 0.120) |
| LIMO (Eckmann et al., 2022) | 99.556 (± 0.228) | 99.511 (± 0.208) | 92.322 (± 4.280) | 99.478 (± 0.247) | 99.689 (± 0.042) |
| GDSS (Jo et al., 2022) | 99.833 (± 0.098) | - | - | - | - |
| MOOD-w/o $P_\phi$ (ours) | 99.827 (± 0.065) | - | - | - | - |
| MOOD-ID (ours) | 98.767 (± 0.335) | 99.613 (± 0.100) | 98.220 (± 0.407) | 97.300 (± 0.119) | 97.860 (± 0.486) |
| MOOD (ours) | 98.860 (± 0.455) | 99.600 (± 0.138) | 98.467 (± 0.322) | 98.693 (± 0.354) | 98.153 (± 0.432) |

Table 12: **Hit ratio (%) results.** The results are the means and the standard deviations of 5 runs. The best results are highlighted in bold.

| Method | Target protein | | | | |
|---|---|---|---|---|---|
| | parp1 | fa7 | 5ht1b | braf | jak2 |
| REINVENT (Olivecrona et al., 2017) | 4.693 (± 1.776) | **1.967** (± 0.661) | **26.047** (± 2.497) | 2.207 (± 0.800) | 5.667 (± 1.067) |
| MORLD (Jeon & Kim, 2020) | 0.047 (± 0.050) | 0.007 (± 0.013) | 0.893 (± 0.758) | 0.047 (± 0.040) | 0.227 (± 0.118) |
| HierVAE (Jin et al., 2020a) | 1.180 (± 0.182) | 0.033 (± 0.030) | 0.740 (± 0.371) | 0.367 (± 0.187) | 0.487 (± 0.183) |
| FREED (Yang et al., 2021) | 4.860 (± 1.415) | 1.487 (± 0.242) | 14.227 (± 5.116) | 2.707 (± 0.721) | 6.067 (± 0.790) |
| FREED-QS | 5.960 (± 0.902) | 1.687 (± 0.177) | 23.140 (± 2.422) | 3.880 (± 0.623) | 7.653 (± 1.373) |
| LIMO (Eckmann et al., 2022) | 0.456 (± 0.057) | 0.044 (± 0.016) | 1.200 (± 0.178) | 0.278 (± 0.134) | 0.711 (± 0.329) |
| GDSS (Jo et al., 2022) | 2.367 (± 0.316) | 0.467 (± 0.112) | 6.267 (± 0.287) | 0.300 (± 0.198) | 1.367 (± 0.258) |
| MOOD-w/o $P_\phi$ (ours) | 2.360 (± 0.234) | 0.480 (± 0.096) | 9.907 (± 0.234) | 0.627 (± 0.132) | 2.780 (± 0.280) |
| MOOD-ID (ours) | 3.860 (± 0.177) | 0.587 (± 0.153) | 15.393 (± 0.567) | 2.860 (± 0.223) | 5.073 (± 0.437) |
| MOOD (ours) | **7.260** (± 0.764) | 0.787 (± 0.128) | 21.427 (± 0.502) | **5.913** (± 0.311) | **10.367** (± 0.616) |

Table 13: **Top 5% docking score results.** The results are the means and the standard deviations of 5 runs. The best results are highlighted in bold.

| Method | Target protein | | | | |
|---|---|---|---|---|---|
| | parp1 | fa7 | 5ht1b | braf | jak2 |
| REINVENT (Olivecrona et al., 2017) | -10.447 (± 0.170) | **-8.510** (± 0.111) | -10.474 (± 0.100) | -10.363 (± 0.136) | -9.565 (± 0.077) |
| MORLD (Jeon & Kim, 2020) | -7.580 (± 0.275) | -6.293 (± 0.167) | -7.877 (± 0.663) | -8.081 (± 0.360) | -7.833 (± 0.135) |
| HierVAE (Jin et al., 2020a) | -9.581 (± 0.195) | -6.842 (± 0.200) | -8.178 (± 0.225) | -9.029 (± 0.270) | -8.347 (± 0.134) |
| FREED (Yang et al., 2021) | -10.607 (± 0.186) | -8.440 (± 0.055) | -10.627 (± 0.283) | -10.493 (± 0.147) | -9.772 (± 0.097) |
| FREED-QS | -10.709 (± 0.100) | -8.475 (± 0.040) | -10.830 (± 0.144) | -10.702 (± 0.074) | -9.849 (± 0.069) |
| LIMO (Eckmann et al., 2022) | -8.986 (± 0.222) | -6.771 (± 0.147) | -8.447 (± 0.052) | -9.048 (± 0.319) | -8.449 (± 0.274) |
| GDSS (Jo et al., 2022) | -10.095 (± 0.031) | -7.921 (± 0.036) | -9.619 (± 0.082) | -9.447 (± 0.054) | -9.027 (± 0.071) |
| MOOD-w/o $P_\phi$ (ours) | -10.155 (± 0.037) | -8.021 (± 0.051) | -9.949 (± 0.064) | -9.769 (± 0.040) | -9.350 (± 0.045) |
| MOOD-ID (ours) | -10.488 (± 0.052) | -8.081 (± 0.041) | -10.602 (± 0.056) | -10.581 (± 0.060) | -9.695 (± 0.054) |
| MOOD (ours) | **-10.898** (± 0.117) | -8.229 (± 0.070) | **-11.194** (± 0.034) | **-11.135** (± 0.037) | **-10.194** (± 0.059) |

Table 14: **Property optimization results** against the target protein parp1 with various values of $\lambda$. The results are the means and the standard deviations of 3 runs. The best results are highlighted in bold.

| $\lambda$ | Novelty (%) | Novel hit ratio (%) | Novel top 5% DS (kcal/mol) |
|---|---|---|---|
| 0.03 | 81.867 ($\pm$ 2.407) | 5.944 ($\pm$ 0.735) | -10.804 ($\pm$ 0.061) |
| 0.04 | 84.180 ($\pm$ 2.123) | **7.017** ($\pm$ 0.428) | **-10.865** ($\pm$ 0.113) |
| 0.05 | **85.467** ($\pm$ 0.694) | 6.444 ($\pm$ 0.457) | -10.803 ($\pm$ 0.086) |

Table 15: **Dockstring F2 task results.** The results are the means and the standard deviations of 3 runs. The best results are highlighted in bold.

| Method | Score |
|---|---|
| FREED (Yang et al., 2021) | -3.391 ($\pm$ 0.209) |
| LIMO (Eckmann et al., 2022) | -0.318 ($\pm$ 0.102) |
| MOOD (ours) | **-3.920** ($\pm$ 0.021) |

To see the effect of $\lambda$ on the property optimization task, we additionally provide the property optimization results with various values of $\lambda$ in Table 14, where $\lambda = 0.04$ is the setting used in the main experiments. As shown in the table, novelty increases as the value of $\lambda$ increases as in the novel molecule generation task. However, the results of $\lambda = 0.03$ and $\lambda = 0.05$ are worse than that of $\lambda = 0.04$ in terms of novel hit ratio and novel top 5% DS, since molecules that deviate from the training distribution naturally tend to be low-quality, and balancing the effect of the OOD control and the property gradient is important to produce novel and high-quality molecules.

We additionally report the results of the F2 task of the Dockstring benchmark (García-Ortegón et al., 2022) in Table 15. The score of the task is calculated as follows:

$$\text{DS (w.r.t. target protein F2)} + 10 \times (1 - \text{QED}). \tag{23}$$

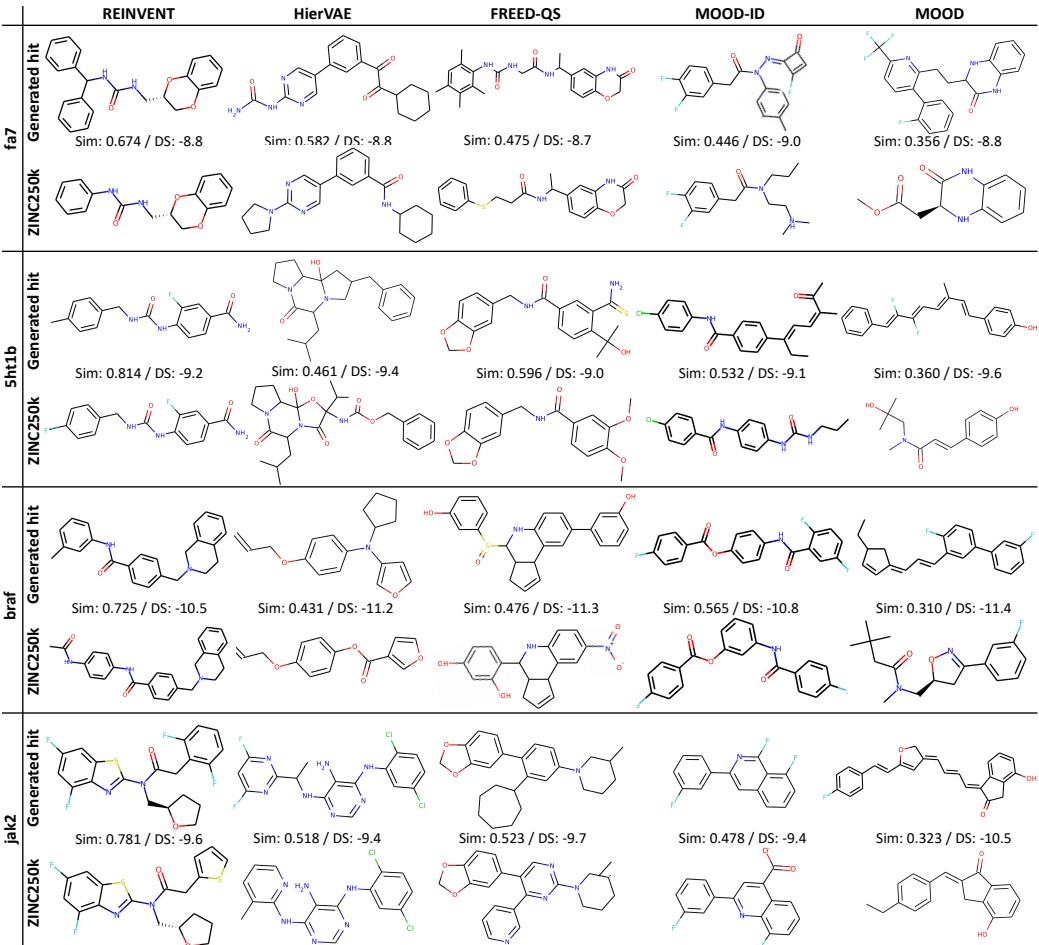

Figure 8: **Generated hit molecules and the corresponding molecules from the ZINC250k dataset of the highest similarity.** The similarity and docking score (kcal/mol) are provided at the bottom of each generated hit.

## D.3 ADDITIONAL MOLECULE SAMPLES

We additionally provide samples of the generated molecules and the training molecules that are maximally similar to those molecules in Figure 8. As shown in the figure, while the molecules generated by the baselines exhibit high Tanimoto similarity, the molecules generated by MOOD do not share big substructures with the training molecules, even in the maximally similar one.

