# OpenReview forum: "Exploring Chemical Space with Score-based Out-of-distribution Generation"
_ICLR.cc/2023/Conference — Submitted to ICLR 2023_

### Official Review · Reviewer_xSWJ · 2022-10-21

**Confidence:** 5
**Correctness:** 4
**Technical Novelty And Significance:** 3
**Empirical Novelty And Significance:** 4
**Recommendation:** 8

**Clarity, Quality, Novelty And Reproducibility:**

The paper is well written, and the analyses performed are supportive of good science. The main novelty is to add a term to conditional generation in reverse diffusion in order to increase the likelihood of generating OOD samples, and showing that it can be used in conjunction with property gradients to generate conditional samples that are somewhat OOD. The authors have included their codebase and it looks comprehensive.

**Strength And Weaknesses:**

Overall, I like the idea of controlling OOD-ness seems somewhat novel, and I think the authors do a good job of demonstrating that by controlling $\lambda$, the OOD-ness can be adjusted. Their analysis of OOD-ness using FCD and MMD (with associated dimensionality-reduction plots) is convincing to me. I also think the combination of OOD-ness with conditional optimization (i.e. the “property gradient”) is a natural next step which the authors address and show results for.
Below are some things I believe could be stronger:
### Property prediction is limited to a single, manually crafted objective
In order to show property prediction, the analysis focuses entirely on a single objective, which combines DS, QED, and SA scores into a single scalar (using the product). It is not clear why this is the chosen objective. The degree of manual tuning in this objective might suggest a lack of robustness in the results. For example, why not use an objective which is the sum of these three scores (with adjustable weightings)?
It would be more convincing if we could see three more distinct property models, each one trained to predict DS, QED, or SA separately. Since the authors are using classifier guidance, the score model would not need to be retrained. Showing that MOOD can then optimize for OOD molecules which show better DS, QED, or SA scores separately would greatly increase the confidence that the method is working and is not limited to a specific, manually crafted objective.
### OOD molecules may generate large numbers of “garbage” molecules
In order to evaluate how well the model generates molecules which satisfy the property objective, the analysis seems to focus entirely on “novel hit ratio” and  “novel top 5% docking score”, which are defined to measure how well the “most OOD” molecules satisfy the property objective. However, these metrics are limited to comparing only the generated molecules which already have a good property score (i.e. the analysis is limited already to molecules which pass certain thresholds). Thus, it is very possible (if not likely) that many molecules generated by this method (not just OOD molecules, but also when $\lambda$ is set to 0) are “garbage” molecules which do not actually correspond to any bioactivity.
It would be much more convincing that this method works, if the authors could compare these property values (i.e. DS, QED, and SA) of all generated molecules (not just those that pass DS/QED/SA thresholds) between their methods (MOOD/MOOD-ID/MOOD-w/o-P) and a couple of benchmarks.
### Effect of $\lambda$ on OOD-ness
It would be nice to also see how the magnitude of $\lambda$ can affect OOD-ness; for example, if lambda is increased further, does it also increase OOD-ness?
### Smaller details
- How is the UMAP in Fig. 4 generated?
- In Fig. 6, it is not clear what the relationship is between the ZINC250k molecules shown and the MOOD-generated molecules: are they the closest ZINC250k molecules based on MMD? Or something else?
- It is not clear how QED and SA scores were computed
- The property-prediction conditional score in Eq. 9 depends on $\lambda$, but it seems (under the last section of Section 3) that the authors don’t actually use lambda in the property-prediction score model. Although this makes some empirical sense, it conflicts with the derivations in Eq. 7 – 9. It would be more clear to have Eq. 7 – 9 assume conditional independence between OOD-ness and property labels to begin with.
- In the sentence following Eq. 13, it should explain what SA is (in structural biology, SA could also refer to solvent accessibility)

**Summary Of The Paper:**

The goal of the paper is to extend molecule generation using diffusion models to include a way to control how “out-of-distribution” generated molecules should be. In order to do this, the authors propose adding a term to the reverse-diffusion score (using an SDE formulation) along with a manually tuned hyperparameter $\lambda$ which controls “OOD-ness”. The authors demonstrate that their method is capable to generating molecules which are unique from the training set, yet still likely have realistic bioactive properties using some computed proxy metrics (i.e. docking score, drug likelihood score, and synthetic accessibility score).

**Summary Of The Review:**

Overall, the paper demonstrates a novel method to control the OOD-ness of generating samples from a generative diffusion model. Their analyses demonstrate convincingly that their method can be used to generate samples that are different from the training set. They also show that within these OOD molecules, there can be found molecules which are expected to have certain bioactive properties.

I find the main weaknesses to be that the paper only focuses on a single computational proxy for bioactivity (which arbitrarily combines a few metrics into a single optimization goal). Thus, the method may not be very robust: it might not work if we want to optimize for a different set of molecular properties. Additionally, there are no analyses that show how many of the generated molecules (OOD or otherwise) are reasonable. It is very possible that the vast majority of molecules generated are garbage.

Together, I think that the contribution of OOD control is interesting and this paper is a good proof of concept, despite its missing analyses that could really show its usefulness in more applications.

---

> ### Author Response · Authors · 2022-11-12
> **Initial Response (2/2)**
>
> **Question 3:**
> It would be nice to also see how the magnitude of $\lambda$ can affect OOD-ness; for example, if $\lambda$ is increased further, does it also increase OOD-ness?
>
> **Answer 3:**
> The effect of $\lambda$ on OOD-ness is shown in Figure 3 (for the ZINC250k dataset) and Table 4 of Appendix (for the QM9 dataset). To see the effect of $\lambda$ on the property optimization task, here we provide the property optimization results with various values of $\lambda$, where $\lambda=0.04$ is our original setting.
>
>
> **Table: Property optimization results with target protein parp1.**
> | $\lambda$ | Novelty (%) | Novel hit ratio (%) | Novel top 5% DS (kcal/mol) |
> | --- | --- | --- | --- |
> | 0.03 | 81.867 | 5.944 | -10.804 |
> | 0.04 | 84.180 | 7.017 | -10.865 |
> | 0.05 | 85.467 | 6.444 | -10.803 |
>
> As shown in the table, novelty increases as the value of $\lambda$ increases as in the novel molecule generation task. However, the results of $\lambda=0.03$ and $\lambda=0.05$ are worse than that of $\lambda=0.04$ in terms of novel hit ratio and novel top 5% DS. Intuitively, molecules that deviate from the training distribution naturally tend to be low-quality, and balancing the effect of the OOD control and the property gradient is important to produce novel and high-quality molecules. Empirically we found $\lambda=0.04$ performs reasonably well throughout the target proteins. We thank you for the suggestion and included the table as Table 14 in Section D.2 of Appendix in the revised paper.
>
> ---
>
> **Question 4:**
>
> 1. How is the UMAP in Fig. 4 generated?
> 2. In Fig. 6, it is not clear what the relationship is between the ZINC250k molecules shown and the MOOD-generated molecules: are they the closest ZINC250k molecules based on MMD? Or something else?
> 3. It is not clear how QED and SA scores were computed.
> 4. The property-prediction conditional score in Eq. 9 depends on λ, but it seems (under the last section of Section 3) that the authors don’t actually use lambda in the property-prediction score model. Although this makes some empirical sense, it conflicts with the derivations in Eq. 7 – 9. It would be more clear to have Eq. 7 – 9 assume conditional independence between OOD-ness and property labels to begin with.
> 5. In the sentence following Eq. 13, it should explain what SA is (in structural biology, SA could also refer to solvent accessibility)
>
> **Answer 4:**
>
> 1. To produce Figure 4, we randomly selected 5,000 molecules from the ZINC250k dataset and 3,000 molecules from REINVENT, FREED-QS, MOOD-ID, and MOOD, respectively. Then, ChemNet [3] activations are computed from the molecules and together visualized by the UMAP library [4]. We included the details of the UMAP visualization in Section C.2 of Appendix.
> 2. As described in the main text, they are the top 0.01\% ZINC250k molecules in terms of binding affinity to parp1. We apologize for omitting the detailed caption in Figure 6 because of the limited space, and this is clarified in the revised paper.
> 3. Following a lot of previous works including all the baselines in the paper, QED and SA scores were computed using the RDKit library [5]. We included the explanation in Section C.3 of Appendix.
> 4. Actually, $P_{\phi}$ can predict the target property without the information of $\lambda$ since chemical properties are entirely determined by molecules themselves, i.e., $P_\phi(\mathbf{G}_t, \lambda)=P_\phi(\mathbf{G}_t)$. We clarified this in the revised paper.
> 5. Thank you for the kind suggestion. We omitted it since we already wrote “synthetic accessibility (SA)” in Introduction, but it will be helpful for readers to include it also in Experiments. We included it in the revised paper.
> ---
>
> **References**
>
> [1] Daria Mochly-Rosen and Kevin Grimes. A practical guide to drug development in academia. Springer Science & Business Media, 2014.
>
> [2] Miguel García-Ortegón et al. Dockstring: easy molecular docking yields better benchmarks for ligand design. Journal of chemical information and modeling 62.15 (2022): 3486-3502.
>
> [3] Kristina Preuer et al. Fréchet chemnet distance: a metric for generative models for molecules in drug discovery. Journal of chemical information and modeling, 58(9):1736–1741, 2018.
>
> [4] Leland McInnes and John Healy. UMAP: uniform manifold approximation and projection for dimension reduction. arXiv:1802.03426, 2018.
>
> [5] Greg Landrum et al. Rdkit: Open-source cheminformatics software. URL http://www.rdkit.org/, https://github.com/rdkit/rdkit, 2016.

---

> > ### Comment · Reviewer_xSWJ · 2022-12-08
> > **Response to initial comments**
> >
> > ### Manually crafted objective
> >
> > Although I agree that manually crafted multi-objective goals are important in practice, both single-objective and multi-objective optimization are important. For example, I might already have a population of molecules which satisfy many constraints, and now I only wish to optimize for a single new property. Thank you for including a second analysis of properties to optimize over. This definitely helps, but the manually crafted nature of the objective still worries me. Would this method still work if you tried to optimize for only DS or only SA, for example? If the method truly works, it should allow us to generate molecules with a good DS or SA score, not just some specific objectives which include hand-selected constants (like DS + (10 * (1 - QED)); why 10?)
> >
> > ### Garbage molecules
> >
> > I see, thank you for clarifying!
> >
> > ### Effect of $\lambda$
> >
> > This is good to see, thank you for including it and with the explanation of non-monotonicity.

---

> ### Author Response · Authors · 2022-11-12
> **Initial Response (1/2)**
>
> We sincerely thank you for your constructive and helpful comments. We appreciate your positive and confident comments that both the technical and empirical novelty are significant, the contribution of the proposed OOD control is interesting, the analyses are convincing, and the claims and statements are well-supported. We address all your concerns below.
>
> ---
>
> **Question 1:**
> The degree of manual tuning in the objective might suggest a lack of robustness in the results. Showing that MOOD can then optimize for OOD molecules which show better DS, QED, or SA scores separately would greatly increase the confidence that the method is working and is not limited to a specific, manually crafted objective.
>
> **Answer 1:**
> The objective function $P_\text{obj}$ was carefully designed to reflect the very basic requirements of real-world *de novo* drug discovery problems. (1) Docking score is not just an important but **essential** property in drug discovery as the generated molecules must show high binding affinity to its target protein to be a drug [1]. (2) Quantitative estimate of drug-likeness (QED) and synthetic accessibility (SA) objectives are required to guarantee the basic chemical plausibility, since a molecule should be drug-like (to filter out odd-looking molecules in chemists’ eyes) and synthesizable. Optimizing a single objective is not appropriate for real-world scenarios, for example, optimizing docking score alone will fail to generate valid drugs since large and absurd molecules frequently show very high absolute docking scores, as explained in the Introduction.
> We appreciate the advice that other objective functions or evaluation metrics would strengthen our work, and here we provide the results of the F2 task of the Dockstring benchmark [2]. The score of the task is $\text{DS (w.r.t. target protein F2)} + 10 \times (1 - \text{QED})$, and we generated molecules with $P_\text{obj}=\hat{\text{DS}}\times \text{QED}$. We also included the results in Section D.2 of Appendix in the revised paper.
>
> **Table: Dockstring F2 task results. The results are the average scores of the generated molecules of 3 runs. Lower is better.**
>
> | Method | Dockstring score |
> | --- | --- |
> | FREED | -3.391 |
> | LIMO | -0.318 |
> | MOOD (ours) | **-3.920** |
> ---
>
> **Question 2:**
> “Novel hit ratio” and “novel top 5% docking score” are limited to comparing only the generated molecules which already have a good property score. Thus, it is very possible that many molecules generated by this method are “garbage” molecules which do not actually correspond to any bioactivity. It would be much more convincing that this method works, if the authors could compare these property values of all generated molecules.
>
> **Answer 2:**
> This is a misunderstanding since the evaluation metrics do consider the generated molecules of low-quality. Novel hit ratio is “the fraction of unique hit molecules that have the maximum Tanimoto similarity less than 0.4 with the training molecules” as explained in Section 4.2 of our paper, which is $\frac{\text{number of unique molecules that are hit and novel}}{\text{total number of generated molecules}}$. Therefore, a model that generates a large number of molecules with low property score shows a low novel hit ratio. Likewise, novel top 5% docking score is calculated using the top 5% molecules of the total number of generated molecules (i.e., the top 150 molecules as the models generate 3,000 molecules), not the top 5% of the good-property molecules. We further clarified the explanation of the evaluation metrics in Section C.4 of Appendix in the revised paper.

---

### Official Review · Reviewer_ZWEt · 2022-10-22

**Confidence:** 4
**Correctness:** 1
**Technical Novelty And Significance:** 3
**Empirical Novelty And Significance:** 2
**Recommendation:** 1

**Clarity, Quality, Novelty And Reproducibility:**

**Clarity**: In general I think the clarity of the paper is good, aside from a few details (e.g. the defintion of $y_0$ in section 3). Some of these details are important however.

**Novelty/Originality**: I don't think I know enough about diffusion models to comment on the novelty of this method compared to other diffusion-model methods. It appears that the core equations of MOOD are novel. From the viewpoint of drug-discovery, I don't think this work is particularly original: it is effectively using a generative model to produce molecules which are predicted to have high properties using a property-prediction model, which describes a huge amount of existing work. To me, it is unclear whether using a diffusion model instead of any other method of exploring chemical space (e.g. a genetic algorithm) addresses any shortcomings of existing methods. Therefore my impression is that the overall contribution to drug discovery is not super large here.

**Quality**: to me, the quality of the motivation for the problem and the method were not very high, and the experiments could be improved by reporting different metrics, additional baseline methods, and trying at least one established benchmark. Therefore my overall assessment of the quality is medium-low.

**Reproducibility**: I'm not an expert on diffusion models, but it seems like the key equations for the method are present. The authors also included code.

**Strength And Weaknesses:**

**Strengths**:

- The core idea of using a diffusion process to generate novel molecules is interesting and could potentially be useful
- The experimental results on 5 non-trivial oracles are interesting and encouraging. I appreciate that they didn't use logP/QED/other trivial oracles
- Writing is very clear

**Weaknesses**

- _Problem is not well-motivated_: the authors state in many places that MOOD is designed to overcome the inability of existing methods to explore areas of chemical space away from the training data. While some methods do have this limitation, a huge number of methods do not, for example methods using genetic algorithms [1, 2] or graph enumeration [3]. Put simply, producing molecules which are "different" is **not** an open problem which would benefit from the development of new methods. My guess would be that the authors are not aware of a lot of earlier literature on this topic (most citations are ML papers from the past 5 years). I would like to see the authors correct this, and perhaps have a more nuanced discussion about the pros/cons of existing methods. Roughly, this would probably be that exploratory methods can propose new molecules but many of them are of low quality, while generative models can propose higher-quality molecules but they are generally more similar to the training data. MOOD could then be described in the context of this trade-off.
- _Mathematical motivation for MOOD is not convincing_. My first concern is that the definition of $y_0$ used to arrive at equation 6 is unclear. The description somewhat implied that it was a binary random variable indicating whether or not a given graph $G$ is OOD, but this would imply a distribution on the set {0, 1} not $[0, 1)$. However, it is also described as a hyperparameter, which means it is _not_ a random variable and therefore writing $p(G|y_0=\lambda)$ and $p(y_0=\lambda|G)$ does not make sense. Equation 5 only adds to this confusion, since it is written as a distribution over $y_0$, but the "derivation" in appendix A.1 treats it like a distribution over $G$ (which I think is a mistake). It also seems to not be a valid probability distribution if $p(G) < p_*$, since equation 5 suggests $p(y_0)=0$ (i.e. it would not normalize to 1 and therefore is not a distribution). Overall this path to arrive at equation 6 makes very little sense to me. I think the authors would need to more clearly define the interpretation and role of $y_0$, and fix any errors in the reasoning that follow (for example if $y_0$ is a random variable then the proof in A.1 is not correct). My second concern is that construction of equation 9 is not really explained at all. My guess is that the authors intended to substitute their model for $p(G_t, y_0, y_p)$ into equation 2, but this is not correct, because it neglects the gradient $\nabla_G\log{Z(G)}$, i.e. the gradient of the normalizing constant in equation 8. The authors should clarify this. Finally, in equation 11, setting the coefficients $\alpha$ to values which depend on the score seems like it would have implications for the correctness of the reverse-time SDE, but this is not discussed (I could be wrong about this though). **Overall**: the key equations of MOOD do not seem to have a principled justification.
- _Missing important baselines_: the baseline methods used by the authors are all methods which tend not to produce OOD molecules. While the inclusion of these baselines is good, the authors should of course include baselines which do not suffer from these problems (see discussion above). The open-source methods from [4] could be a good starting point, for example Graph GA. These methods will almost certainly explore as well as MOOD or better, and could plausibly also produce higher-scoring molecules in section 4.2
- _Issues with metrics in section 4.2_: the novel hit ratio and novel top 5% docking score are both normalized by the number of unique (hit) molecules, which can be problematic (assuming I have understood them correctly; if not then the authors should explain their metrics more clearly). For example, out of 3000 molecules, method A produces just 2 unique molecules (one novel hit and one poor molecule), it's novel hit ratio would be 100%, while method B producing 500 novel hits, 500 similar hits, and 2000 poor molecules would have a novel hit ratio of 50%. I think method B would be superior in this case, while the author's metrics would favour method A. Novel top 5% docking score suffers from similar problems. I think the authors should either report standard metrics, or continue to sample from all methods until a certain number of unique hit molecules are produced, which would fix the normalization problem.
- _Lack of experiments on standard benchmarks_: while I support the authors creation of a new, reasonable molecular optimization benchmark, the problem with just evaluating MOOD on a novel task is that the general difficulty and idiosyncrocies of this task are not understood, so it is unclear how much of the method's success is "general" and how much is due to being particularly well-suited for these specific tasks. I think that evaluating MOOD on at least one standard benchmark would be useful. One option would be to use the benchmark from PMO [4], which includes a variety of semi-toy functions such as GuacaMol. These are not as realistic as docking, but have also been tried with many more baselines, which would help contextualize the performance of MOOD. A second option using docking would be dockstring [5] which contains several standardized docking benchmarks.

[1] Nigam, AkshatKumar, et al. "Beyond generative models: superfast traversal, optimization, novelty, exploration and discovery (STONED) algorithm for molecules using SELFIES." Chemical science 12.20 (2021): 7079-7090.

[2] Jensen, Jan H. "A graph-based genetic algorithm and generative model/Monte Carlo tree search for the exploration of chemical space." Chemical science 10.12 (2019): 3567-3572.

[3] Ruddigkeit, Lars, et al. "Enumeration of 166 billion organic small molecules in the chemical universe database GDB-17." Journal of chemical information and modeling 52.11 (2012): 2864-2875.

[4] Gao, Wenhao, et al. "Sample efficiency matters: a benchmark for practical molecular optimization." arXiv preprint arXiv:2206.12411 (2022).

[5] García-Ortegón, Miguel, et al. "DOCKSTRING: easy molecular docking yields better benchmarks for ligand design." Journal of chemical information and modeling 62.15 (2022): 3486-3502.

**Summary Of The Paper:**

This paper proposes MOOD (Molecular out of distribution diffusion), a method using diffusion models to generate novel molecules which differ frrom training molecules and potentially have desirable properties. The method essentially modifies the reverse-time SDE equation to reduce the coefficient of the $\nabla \log{p}$ (score) term and add a term involving a property predictor. Experimentally they demonstrate that MOOD produces molecules which are:
1. Different than the training set
2. Possess desirable properties using an oracle based on molecular docking

**Summary Of The Review:**

This paper proposes a potentially interesting idea to a problem which I don't think is actually a problem; instead it is just a shortocming of several popular models in ML over the past 5 years. The derivation of the method itself is unclear, which makes the method seem not very principled (EDIT: further discussions with the authors make me think that several key equations are incorrect due to a mis-use of Bayes rule, so I lowered my score). I do not find the experimental results convincing, particularly because of the omission of a large class of important baseline methods. Together, these shortcomings imply that the paper should not be accepted.

---

> ### Author Response · Authors · 2022-11-12
> **Initial Response (3/3)**
>
> **Question 3:**
> The novel hit ratio and novel top 5% docking score are both normalized by the number of unique (hit) molecules, which can be problematic.
>
> **Answer 3:**
> This is a misunderstanding about the metrics. As explained in Section 4.2 of our paper, novel hit ratio is “the fraction of unique hit molecules that have the maximum Tanimoto similarity less than 0.4 with the training molecules”, which is $\frac{\text{number of unique molecules that are hit and novel}}{\text{total number of generated molecules}}$. Likewise, novel top 5% docking score is calculated using the top 5% of the total generated molecules (i.e., the top 150 molecules as the models generate 3,000 molecules), not the top 5% of the unique molecules. Also, the uniqueness results are included in Table 11 of Appendix, which shows our MOOD achieves near-perfect uniqueness. We thank you for the kind suggestion and further clarified the explanation of the evaluation metrics in Section C.4 of Appendix in the revised paper.
>
> ---
>
> **Question 4:**
> Evaluating MOOD on at least one standard benchmark would be useful, e.g., Dockstring.
>
> **Answer 4:**
> We sincerely thank you for the suggestion. Docking score is not just an important but **essential** property in drug discovery as the generated molecule must show high binding affinity to its target protein to be a drug [1]. Most works on deep drug discovery and most standard benchmarks including GuacaMol focus on the more elementary stage, which aims to generate chemically plausible molecules. The main purpose of our paper is to build a generative model that can solve practical *de novo* drug discovery problems, so the focus is different. Also, the molecule design tasks in Dockstring are basically very similar to our setting except they do not consider novelty, as the evaluation metric is the combination of the docking score and QED score [2]. This is a valuable suggestion and here we provide results of the F2 task of the Dockstring benchmark. We also included the results in Section D.2 of Appendix in the revised paper.
>
> **Table: Dockstring F2 task results. The results are the average scores of the generated molecules of 3 runs. Lower is better.**
> | Method | Dockstring score |
> | --- | --- |
> | FREED | -3.391 |
> | LIMO | -0.318 |
> | MOOD (ours) | **-3.920** |
>
> ---
>
> **References**
>
> [1] Daria Mochly-Rosen and Kevin Grimes. A practical guide to drug development in academia. Springer Science & Business Media, 2014.
>
> [2] Miguel García-Ortegón et al. Dockstring: easy molecular docking yields better benchmarks for ligand design. Journal of chemical information and modeling 62.15 (2022): 3486-3502.
>
> [3] Song et al. Score-Based Generative Modeling through Stochastic Differential Equations. ICLR, 2021.

---

> ### Author Response · Authors · 2022-11-12
> **Initial Response (2/3)**
>
> **Question 2-1:**
> About the definition of $y_o$ and the probability $p(G|y_o=λ)$ and $p(y_o=λ|G)$.
>
> **Answer 2-1:**
> Note that $y_o$ is a random variable that represents the condition for OOD-ness which is quantified as a scalar value $\lambda$ in our paper, and a large value of $y_o$ represents it deviates from the training data. This notation is similar to that of [3] which is widely used to represent conditions such as class. Further, $p_t(G|y_o=\lambda)$ is the probability of graph $G$ with OOD-ness $\lambda$ at time $t$, and $p_t(y_o=\lambda|G)$ is the probability of OOD-ness $\lambda$ for the given graph $G$ at time $t$.
>
> **Question 2-2:**
> Equation 5 is written as a distribution over $y_o$, but the "derivation" in appendix A.1 treats it like a distribution over $G$ (which I think is a mistake). It also seems to not be a valid probability distribution if $p(G)<p^∗$, since equation 5 suggests $p(y_o)=0$ (i.e. it would not normalize to 1 and therefore is not a distribution).
>
> **Answer 2-2:**
> Equation 5 describes that the probability of OOD-ness (where OOD-ness is quantified as $\lambda$) for the given graph $G$ at time $t$ is proportional to the negative exponent of the density $p_t(G_t)$. To show that there exists a distribution proportional to $p_t(G_t)^{-\sqrt{\lambda}}$, we provided the derivation in A.1. With a sufficiently small threshold, the probability in Equation 5 is valid.
>
> **Question 2-3:**
> Construction of equation 9 is not correct, because it neglects the gradient $\nabla_G \log⁡Z(G)$, i.e. the gradient of the normalizing constant in equation 8.
>
> **Answer 2-3:**
> Equation 9 is the result of using Equation 5 and Equation 8 as explained in the paper. When computing the gradient of Equation 8 with respect to the data $G$, the gradient $\nabla_{G}\log Z(G)$ can be neglected as the normalizing constant $Z$ is independent of $G$ since it is defined as $Z=\int_{G}e^{\alpha_t P_{\phi}(G,\lambda)}$.
>
> **Question 2-4:**
> Finally, in equation 11, setting the coefficients $\alpha$ to values which depend on the score seems like it would have implications for the correctness of the reverse-time SDE, but this is not discussed (I could be wrong about this though).
>
> **Answer 2-4:**
> The coefficients $\alpha_1$ and $\alpha_2$ balance the effect of OOD control and the property gradient throughout the diffusion process as explained in Section 3.2, and are not related to the correction of SDEs.
>
> We believe that the derivations of the OOD-controlled diffusion process and the conditional generation process are well-defined and properly justified, and we have extensively shown that the proposed framework is effective for novel molecule generation that satisfies the given constraints.

---

> ### Author Response · Authors · 2022-11-12
> **Initial Response (1/3)**
>
> We sincerely thank you for your constructive comments. We appreciate your positive comments that the core idea is interesting and novel, the selection of oracles in the experiments are thorough and non-trivial, and the writing is very clear. We address all your concerns below.
>
> ---
>
> **Question 1:**
> The authors state in many places that MOOD is designed to overcome the inability of existing methods to explore areas of chemical space away from the training data. While some methods do have this limitation, a huge number of methods do not.
>
> The baseline methods used by the authors are all methods which tend not to produce OOD molecules. While the inclusion of these baselines is good, the authors should of course include baselines which do not suffer from these problems, for example Graph GA.
>
> **Answer 1:**
> The goal of our paper is to overcome the limited exploration problem of **generative models** on molecule generation, which mostly rely on distribution learning or subgraph assembling (i.e., those utilize molecular motif vocabularies) to generate high-quality molecules. Also, we compared MOOD with the baselines that do not utilize training data or a motif vocabulary, e.g., MORLD (originally in Table 1 and 2; see Table 5 and 6 in the revised paper) and GCPN, and showed that they fail to generate high-quality drug candidates. Here we additionally provide the results of Graph GA.
>
> **Table: Novel hit ratio (%) results.**
> | Method | parp1 | fa7 | 5ht1b | braf | jak2 |
> | --- | --- | --- | --- | --- | --- |
> | Graph GA | 4.811 | 0.422 | 7.011 | 3.767 | 7.644 |
> | FREED-QS | 4.627 | **1.332** | 16.767 | 2.940 | 5.800 |
> | LIMO | 0.455 | 0.044 | 1.189 | 0.278 | 0.689 |
> | MOOD (ours) | **7.017** | 0.733 | **18.673** | **5.240** | **9.200** |
>
> **Table: Novel hit ratio (%) results with the Tanimoto similarity constraint of 0.3.**
> | Method | parp1 | fa7 | 5ht1b | braf | jak2 |
> | --- | --- | --- | --- | --- | --- |
> | Graph GA | 0.311 | 0.011 | 0.644 | 0.267 | 0.778 |
> | FREED-QS | 0.547 | 0.220 | 1.633 | 0.260 | 0.773 |
> | LIMO | 0.433 | 0.033 | 0.922 | 0.267 | 0.644 |
> | MOOD (ours) | **4.107** | **0.387** | **10.687** | **3.293** | **5.525** |
>
> As shown in the results, our proposed MOOD outperforms Graph GA in generating novel and high-quality molecules. The performance gap increases with the harsher novelty condition (0.3), showing that MOOD is indeed very effective in generating drug candidates that are both novel and high-quality and superior in *de novo* drug discovery tasks. We appreciate your kind suggestion, and we clarified that the limited exploration problem is that of distribution learning models or fragment-based models in Related Work and added the above results in the revised paper. The results of the baselines that are not based on distribution learning or a fragment vocabulary are provided as Table 5 and 6.

---

> > ### Comment · Reviewer_ZWEt · 2022-11-19
> > **Response to authors**
> >
> > Thank you for responding to my review. I am very sorry that I did not respond to you earlier: I did not get an email from OpenReview notifying me that you had written a rebuttal, which happeded for all the other papers I reviewed. I assumed you had just not made any comments.
> >
> > I will briefly respond to your comments (all 3 responses)
> >
> > **Question 1**: problem motivation
> >
> > I would argue that methods like GraphGA can easily be cast as [implicit] generative models, for example the distribution realized by running the algorithm for a fixed number of steps. The new results are encouraging though and I do think that they strengthen the paper.
> >
> > **Question 2**: potential problems with math
> >
> > I did not find your response very convincing. I think defining $y_o$ as "OOD-ness" is vague. Is just a subjective value? Is it related to log probability under some model? Does it have a mathematical definition? I looked at [3] and did not find a corresponding variable.
> >
> > Regarding equation 5: the derivation in appendix A.1 normalizes w.r.t. G, when the distribution is over $y_o$. Therefore I believe it is still not valid. I also believe that since equation 8 is a distribution over $y_p$ the normalizing constant depends on $G,y_o$ and therefore $\nabla_G \log{Z} \neq0$. I don't mean to be rude, but I think the authors are either confused about what are the random variables in each equation, or they have written the equations using notation which is similar to standard Bayesian notation but means something different. I think it would be a helpful exercise to write out the full joint probability distribution of all variables in this problem, which may provide insight as to where the problems lie (if there are problems).
> >
> > **Question 3**: metrics
> >
> > You are right, I misunderstood this metric. I still think that normalizing by the total number of generated molecules is an unrealistic metric, since a property predictor could be use to "enrich" other distributions, so this comparison is maybe not very fair. But it is better than I originally thought!!
> >
> > **Question 4**: another benchmark
> >
> > These are good results. Thanks for running this experiment!
> >
> > **Overall**: although this response has addressed some of my concerns, my main concern is that the method is not very principled/well-motivated/well-defined, and this was not really resolved by the authors' response to question 2 above. In general I think that methods which are not principled need to meet an extra-high empirical bar to be accepted, and I don't think that MOOD meets this (for example I think it would need to compare against genetic algorithms with property predictors). Therefore I will not change my score at this time. I am sorry that because I am responding so late the authors probably won't be able to engage with my comments, which is my bad (but also arguably OpenReview's fault for not emailing me).

---

> > > ### Author Response · Authors · 2022-11-21
> > > **Response to Reviewer ZWEt**
> > >
> > > We sincerely thank you for replying to our response and acknowledging our efforts in the new experimental results. We address your concerns below. Please note that we clearly separate the notations of random variable and scalar values in our paper by using bold symbols, for example $\mathbf{G}_t$ is a random variable while $\lambda$ is a scalar.
> > >
> > > ---
> > >
> > > **Question 1-1:**
> > > I think defining $y_o$  as "OOD-ness" is vague. Is it related to log probability under some model and does it have a mathematical definition? I looked at [3] and did not find a corresponding variable.
> > >
> > > **Answer 1-1:**
> > > $y_o$ is a random variable that represents a “condition” OOD-ness which is quantified by a scalar value $\lambda$, and **we clearly define the conditional probability of the condition $\mathbf{y}_o=\lambda$ for given random variable $\mathbf{G}_t$ in equation 5**.
> > > Our initial response regarding reference [3] was pointing to your misunderstanding of the notation of the random variable in the conditional probabilities which we similarly follow. Please refer to equation 14 and the equations in section I of the appendix in [3] for similar notations.
> > >
> > > ---
> > >
> > > **Question 1-2:**
> > > Regarding equation 5, the derivation in appendix A.1 normalizes w.r.t. G, when the distribution is over $y_o$, thus I believe it is still not valid.
> > >
> > > **Answer 1-2:**
> > > Derivation in appendix A.1 shows that there exists a distribution proportional to $p_t(\mathbf{G}_t)^{-\sqrt{\lambda}}$, where $\lambda$ is a scalar value. Thus **this guarantees that we can define the distribution $p_t(\mathbf{y}_0=\lambda|\mathbf{G}_t))$ that is proportional to $p_t(\mathbf{G}_t)^{-\sqrt{\lambda}}$**. Please let us know if anything remains unclear about this point, since this derivation is perfectly valid.
> > >
> > > ---
> > >
> > > **Question 1-3:**
> > > I believe that since equation 8 is a distribution over $y_p$, the normalizing constant depends on $G, y_o$.
> > >
> > > **Answer 1-3:**
> > > **The normalizing constant does not depend on $\mathbf{G}$ as it is the expectation over the support of $\mathbf{G}$**, which is similar to the normalizing constant used in the energy-based models [4]. More specifically, since the definition of the normalizing constant in equation 8 is the expectation of the value $e^{\alpha_t P_{\phi}(G,\lambda)}$ over the support of the random variable $\mathbf{G}$, it cannot depend on the random variable $\mathbf{G}$. Therefore, $\nabla_{\mathbf{G}}\log Z=0$.  We hope this clears the misunderstanding regarding the derivative of the normalizing constants.
> > >
> > > ---
> > >
> > > **Question 2:**
> > > I think that methods which are not principled need to meet an extra-high empirical bar to be accepted, and I don't think that MOOD meets this.
> > >
> > > **Answer 2:**
> > > Could you please provide us a more concrete reason why you think that the method is not principled? If this comment was regarding questions 1-1 through 1-3, please refer to our answers where we did our best to correct your misunderstandings. As described above, we strongly believe that **our theoretical grounds are solid.**
> > >
> > > Moreover, as described in our initial response, we emphasize that **the goal of our paper is to overcome the limited exploration problem of generative models for molecule generation, and thus our direct competitors are generative models**, and we strongly believe that we made faithful comparisons against them, including VAE-based models (JTVAE, HierVAE, and LIMO), flow-based models (GraphAF and GraphDF), and a score-based model (GDSS). In addition, we included comparisons with **non-generative models** such as **RL-based models (REINVENT, GCPN, MORLD, and FREED)** and further included a **GA-based model (Graph GA) following your initial review**. These models encompass most of the existing strong molecule generation methods, and we experimentally demonstrated that our proposed MOOD outperforms these methods in the practical **de novo** drug discovery problems. We strongly believe that our experimental validation was extra thorough, compared to existing works on molecular generation that compare against far fewer baselines.
> > >
> > > ---
> > > [4] Song and Kingma. How to Train Your Energy-Based Models. arXiv, 2021.

---

> > > > ### Comment · Reviewer_ZWEt · 2022-11-25
> > > > **Response to authors about the theoretical motivation for the method**
> > > >
> > > > Thank you for your response and sorry for not responding sooner: once again I did not get an email notification; I only saw this response when I got an email about your response to another reviewer. I appreciate your answers to my questions, but they still did not resolve any of my concerns/misunderstandings. I think there is definitely a misunderstanding between us which I would like to resolve. I will re-visit my issues below:
> > > >
> > > > **Question 1-1**: I think my issue is that I don't understand the generative process. I would understand if $y_o$ is a fixed value and not a random variable, then the generative process is just equation 3. However, you treat $y_o$ as a random variable and use Bayes rule to define $p(G_t)$ in equations 4-5. I interpreted this as implying a prior over $y_o$, which you do not give. If there is no prior, then what is $p(G_t)$ written without reference to $y_o$? Is this a separate, unrelated generative process, or the marginal distribution of $\int p(G_t, y_o) dy_o$? Please try to help me understand this more precisely. Writing out the full probabilistic model would be helpful perhaps.
> > > >
> > > > **Question 1-2**: probability distributions must be non-negative and integrate to 1. Your distribution is $p(y_o=\lambda | G)$, and in appendix A.1 you show that there is a distribution such that $\int p(y_o=\lambda | G) dG = 1$ instead of $\int p(y_o=\lambda | G) d\lambda = 1$. That is why I said it is invalid. Your derivation in appendix A.1 reads as if it were deriving a distribution over $G$, since equation 14 describes a distribution over $G$. I believe it is valid for $G\in H_t$ but otherwise not (since it is 0 for all $\lambda$ and therefore does not integrate to 1). Does this make sense?
> > > >
> > > > **Question 1-3**: the random variable in this equation is $y_p$, so it must satisfy $\int p(y_p|G, y_o) dy_p = 1$. Equation 8 is a valid distribution if and only if $Z=\int p(y_p|G, y_o) dy_p$. Clearly this depends on G. Your answer and equations in the main text suggest that the random variable is $G$. Does this make sense?
> > > >
> > > > **Question 2**: yes, the main reason I think MOOD is not principled is because I do not believe that the theoretical justification is valid. If you can convince me that I am misunderstanding the questions above then I would agree that the method is principled.
> > > >
> > > > I will check this thread more frequently in the future to make sure I don't miss your response.

---

> > > > > ### Author Response · Authors · 2022-11-27
> > > > > **Third Response to Reviewer ZWEt**
> > > > >
> > > > > We thank you for replying to our response. We address your concerns below.
> > > > >
> > > > > ---
> > > > >
> > > > > **Question 1:**
> > > > > I think my issue is that I don't understand the generative process. You treat $\mathbf{y}_o$ as a random variable and use Bayes rule to define $p(\mathbf{G}_t)$ in equations 4-5. I interpreted this as implying a prior over $\mathbf{y}_o$, which you do not give. If there is no prior, then what is $p_t(\mathbf{G}_t)$ written without reference to $\mathbf{y}_o$? Is this a separate, unrelated generative process, or the marginal distribution of $\int p(\mathbf{G}_t, \mathbf{y}_o)\mathrm{d}\mathbf{y}_o$?
> > > > >
> > > > > **Answer 1:**
> > > > > First, we note that the idea of treating the condition (e.g., class condition, unknown dimensions for imputation, and the OOD condition in our work) as a random variable and using Bayes’ rule to define $p(\mathbf{G}_t)$ is very widely used in the conditional generation of score-based models, and we have provided the reference: Equation 14 and section I of the appendix of Song et al. [1], in our previous responses. For more general details of the conditional generation of score-based models, especially general inverse problems, please refer to section 5 and section I of the appendix of Song et al. [1].
> > > > >
> > > > > The conditional reverse-time SDEs in equation 3 of our paper describes the diffusion process from noise to data, and the random variable $\mathbf{G}_t$ follows the marginal distribution $p_t(\mathbf{G}_t|\mathbf{y}_o)$ at time $t$. To generate out-of-distribution samples, we define an OOD condition represented by the random variable $\mathbf{y}_o$, and to generate samples from the unknown distribution $p_t(\mathbf{G}_t|\mathbf{y}_o)$, we used Bayes’ rule as in equation 4. Here, we propose to model the distribution $p_t(\mathbf{y}_o=\lambda|\mathbf{G}_t)$, which is the probability that the molecular graph $\mathbf{G}_t$ satisfies the OOD condition $\mathbf{y}_o=\lambda$, using equation 5.
> > > > >
> > > > > [1] Song et al. Score-Based Generative Modeling through Stochastic Differential Equations. ICLR, 2021.
> > > > >
> > > > > ---
> > > > >
> > > > > **Question 2:**
> > > > > Your derivation in appendix A.1 reads as if it were deriving a distribution over $\mathbf{G}$, since equation 14 describes a distribution over $\mathbf{G}$. I believe it is valid for $\mathbf{G}\in \mathbf{H}_t$ but otherwise not (since it is 0 for all $\lambda$ and therefore does not integrate to 1).
> > > > >
> > > > > **Answer 2:**
> > > > > Although we have answered the same question in our previous responses, we try to answer again more clearly.
> > > > > 1. $p_t(\mathbf{y}_o|\mathbf{G}_t)$ represents the probability that the molecular graph $\mathbf{G}_t$ satisfies the property $\mathbf{y}_o$.
> > > > > 2. Thus we have proposed to model the distribution $p_t(\mathbf{y}_o|\mathbf{G}_t)$ to be proportional to the negative exponent of the density $p_t(\mathbf{G}_t)$.
> > > > > 3. The derivation in appendix A.1 shows that there exists a probability distribution that is proportional to the negative exponent of the density $p_t(\mathbf{G}_t)$. $q(\mathbf{G})=0$ for $\mathbf{G} \notin \mathbf{H}$ does not mean that $q(\mathbf{G})$ is not a probability distribution, since as you said, $q(\mathbf{G})$ can be integrated to $1$ in the domain $\mathbf{H}$ (this is why in equation 15 the integration over $\mathcal{G}$ is converted to the integration over $\mathbf{H}$).
> > > > >
> > > > > We are certain that all of the steps are valid, and we have clearly shown with extensive experiments that our proposed modeling of equation 5 is valid and effective.
> > > > >
> > > > > ---
> > > > >
> > > > > **Question 3:**
> > > > > Equation 8 is a valid distribution if and only if $Z=\int p(\mathbf{y}_p|\mathbf{G},\mathbf{y}_o)\mathrm{d}\mathbf{y}_p$.
> > > > >
> > > > > **Answer 3:**
> > > > > As clearly explained in the paragraph above equation 7, $p_t(\mathbf{y}_p|\mathbf{G}_t,\mathbf{y}_o=\lambda)$ represents the **probability that the molecular graph $\mathbf{G}_t$ satisfies the property $\mathbf{y}_p$**, which we propose to model the probability density using the Boltzmann distribution $p_t(\mathbf{y}_p|\mathbf{G}_t,\mathbf{y}_o=\lambda) = e^{\alpha_t P_\phi(\mathbf{G}_t, \lambda)}/Z_t$. Here $Z_t$ is simply the normalizing constant which is defined as $Z_t = \int e^{\alpha_t P_\phi(\mathbf{G}_t, \lambda)}\mathrm{d}\mathbf{G}_t$ to make the right-hand side of equation 8 a probability distribution. **Therefore, $Z_t$ does not depend on $\mathbf{G}_t$**. If you are not familiar with the Boltzmann distribution, please refer to Liu et al. [2] or other works on energy-based models (EBMs) as **the usage of Boltzmann distribution and its gradient is the key strategy of EBMs**.
> > > > >
> > > > > [2] Liu et al. GraphEBM: Molecular graph generation with energy-based models. ICLR Energy-Based Models Workshop, 2021.

---

> > > > > > ### Comment · Reviewer_ZWEt · 2022-11-27
> > > > > > **Acknowledge response but my concerns are not resolved**
> > > > > >
> > > > > > Thank you once again for responding to me. I think we are not on track to reach an agreement on this, which is frustrating (I sense that you are frustrated). I will summarize what issues I feel are unresolved:
> > > > > >
> > > > > > **Question 1 (the model in general)**: thank you for confirming my suspicion that this method is intended to be Bayesian. I still think the model is not fully specified. For a Bayesian model to be fully specified, you would need to additionally specify a prior $p(y_o)$ in order to fully specify the joint distribution. My understanding is that you have not done this. I tried to ask for this in my previous response but you did not respond to this request specifically as far as I can tell.
> > > > > >
> > > > > > **Question 2 (normalization issue 1)** Thank you for writing out your reasoning specifically. I think your error is in step 3. This line of reasoning does not show that the distribution can be normalized wr.t. $\lambda$, which is what you would need to show to complete this proof. I’m not sure why you think you should be integrating over $G$; this is not what Bayes rule says. Perhaps you can explain why you think we should be normalizing over $G$ instead of $\lambda$?
> > > > > >
> > > > > > **Question 3 (normalization issue 2)** My issue here is that your equation for $p(y_p|G, \lambda)$ does not specify a value of $y_p$ anywhere. This equation should contain a value of $y_p$ unless the density is constant. Bayes rule states that you should integrate wrt $y_p$, **not** $G$. If you believe [2] says otherwise, then I think you misunderstood this work: they are modelling a distribution over $G$ which is why it is the correct choice, but you are modelling a distribution over $y_p$ which is not the same thing. To respond to this concern, perhaps the authors could explain why $y_p$ does not appear in this equation, even though it claims to specify a distribution over $y_p$?
> > > > > >
> > > > > > My impression is that the authors may not fully understand Bayes rule and that this is the source of our disagreement. I asked a colleague for a second opinion and he agreed that this is the best explanation of our disagreement. I am happy to be convinced otherwise (e.g. if the authors provide complete and detailed equations showing how the equations satisfy Bayes rules). However, we’ve now had several iterations of essentially saying the same thing at each other without our positions changing…

---

> > > > > > > ### Author Response · Authors · 2022-11-29
> > > > > > > **Response to Reviewer ZWEt**
> > > > > > >
> > > > > > > We thank you for your response. We hope our responses will resolve the misunderstandings, and it would greatly help our discussion if you could take a look at the reference Song et al. [1] for the general formulation of the conditional score-based models. We address your concerns below.
> > > > > > >
> > > > > > > [1] Song et al. Score-Based Generative Modeling through Stochastic Differential Equations. ICLR, 2021.
> > > > > > >
> > > > > > > ---
> > > > > > >
> > > > > > > Question 1: For a Bayesian model to be fully specified, you would need to additionally specify a prior $p(\mathbf{y}_o)$ in order to fully specify the joint distribution.
> > > > > > >
> > > > > > > Answer 1: The prior of $p_t(\mathbf{y}_o)$ can be specified as $\int p_t(\mathbf{y}_o|\mathbf{G}_t) p_t(\mathbf{G}_t)\mathrm{d}\mathbf{G}_t$, and the prior $p(\mathbf{y}_o)$ itself is meaningless without the graph $\mathbf{G}_t$ given. As we have emphasized in our previous response, the idea of treating the condition (e.g., class condition, unknown dimensions for imputation, and the OOD condition in our work) as a random variable and using Bayes’ rule to define $p(\mathbf{G}_t)$ is widely used [1,2,3].
> > > > > > >
> > > > > > > For example, as described in the reference Song et al. [1], for class-conditional image generation we sample from $p_t(\mathbf{x}_t|\mathbf{y})$ where $\mathbf{x}_t$ is the random variable of the image and $\mathbf{y}$ is the random variable that represents the class condition. The prior $p(\mathbf{y})$ is not important, as the focus is sampling from $p_t(\mathbf{x}_t|\mathbf{y})$.
> > > > > > >
> > > > > > > [1] Song et al. Score-Based Generative Modeling through Stochastic Differential Equations. ICLR, 2021.
> > > > > > >
> > > > > > > [2] Ho and Salimans, Classifier-Free Diffusion Guidance, arxiv preprint 2022
> > > > > > >
> > > > > > > [3] Dhariwal and Nichol, Diffusion Models Beat GANs on Image Synthesis, NeruIPS 2021
> > > > > > >
> > > > > > > ---
> > > > > > >
> > > > > > > Question 2: I’m not sure why you think you should be integrating over $G$; this is not what Bayes rule says. Perhaps you can explain why you think we should be normalizing over $G$ instead of $\lambda$?
> > > > > > >
> > > > > > > Answer 2: As explained in the first sentence of page 5 of our paper, our proposed model of equation 5 induces the distribution proportional to $p_t(\mathbf{G}_t)^{1-\sqrt{\lambda}}$. So in view of the Bayes rule, to show that there exists a distribution $p_t(\mathbf{G}_t|\mathbf{y}_o)$ proportional to $p_t(\mathbf{G}_t)^{1-\sqrt{\lambda}}$, we should be normalizing over $G$, and this is equivalent to the derivation of appendix A.1.
> > > > > > >
> > > > > > > ---
> > > > > > >
> > > > > > > Question 3: My issue here is that your equation for $p(y_p|G,\lambda)$ does not specify a value of $y_p$ anywhere. This equation should contain a value of $y_p$ unless the density is constant. To respond to this concern, perhaps the authors could explain why  does not appear in this equation, even though it claims to specify a distribution over $y_p$?
> > > > > > >
> > > > > > > Answer 3: It is a critical misunderstanding that $y_p$ is not contained in equation 8. $P_{\phi}$ is the property function that incorporates the information of $\mathbf{y}_p$ which represents the property condition, and we estimate the property function with the property prediction network. Therefore the value of $\mathbf{y}_p$ does not have to be constant.

---

> > > > > > > > ### Comment · Reviewer_ZWEt · 2022-11-29
> > > > > > > > **Questions not adequately answered; end of discussion**
> > > > > > > >
> > > > > > > > Thanks for your response. I had read reference [1] a long time ago but it did not fully answer my questions which is why I posted a comment. I have no problem with class-conditional sampling in their case, but your case is slightly different because parts of your model require estimating actual values of probabilities, for example to define the set $H_t$. For question 1, my main question is: what is $p(G_t)$? Is it a separate, unconditional diffusion model?
> > > > > > > >
> > > > > > > > Question 2: I don't understand your answer... the joint distribution would need to be normalized with respect to $G$ and $\lambda$ and you don't show this.
> > > > > > > >
> > > > > > > > Question 3: $y_p$ is only on the left side of equation 8. The relationship between $p_\phi$ and $y_p$ is not clearly stated....
> > > > > > > >
> > > > > > > > I'm sorry to say that I think it is no longer worth my time to engage in this discussion since I have not raised my score after many rounds of back and forth, I find it hard to believe that subsequent discussion will not change anything. I thank the authors for answering my questions, but I think most reviewers would not have engaged this much and that in the future the authors should write longer, more clear answers to the reviewer questions about the model, rather than point reviewers to references and give very brief answers. I still believe your model is not well-specified and that Bayes rule is not satisfied in multiple places. I really encourage you to get a second opinion on the math in your paper. Good luck with whatever the future of this paper may be.

---

> > > > > > > > > ### Author Response · Authors · 2022-12-02
> > > > > > > > > **Response to Reviewer ZWEt**
> > > > > > > > >
> > > > > > > > > **Question 1:**
> > > > > > > > > What is $p(G_t)$? Is it a separate, unconditional diffusion model?
> > > > > > > > >
> > > > > > > > > **Answer 1:**
> > > > > > > > > Yes, $p(\mathbf{G}_t)$ is an unconditional diffusion model, as we explained in equation 2.
> > > > > > > > >
> > > > > > > > > ---
> > > > > > > > >
> > > > > > > > > **Question 2:**
> > > > > > > > > The joint distribution would need to be normalized with respect to $G$ and $\lambda$.
> > > > > > > > >
> > > > > > > > > **Answer 2:**
> > > > > > > > > Our goal is to model the conditional probability $p_t(\mathbf{G}_t|\mathbf{y}_o)$ so that we could generate graphs with the OOD condition $\mathbf{y}_o$, and thereby proved the existence of such a distribution by normalizing with respect to $\mathbf{G}$.
> > > > > > > > >
> > > > > > > > > ---
> > > > > > > > >
> > > > > > > > > **Question 3:**
> > > > > > > > > $y_p$ is only on the left side of equation 8. The relationship between $P_\phi$ and $y_p$ is not clearly stated.
> > > > > > > > >
> > > > > > > > > **Answer 3:**
> > > > > > > > > In our paper, we propose to model the property distribution using the Boltzmann distribution as $p_t(\mathbf{y}_p | \mathbf{G}_t, \mathbf{y}_o=\lambda)=e^{\alpha_t P_\phi(\mathbf{G}_t,\lambda)}/Z_t$. $\mathbf{y}_p$ is an intermediate variable that we set to help understanding, and the relationship between $\mathbf{y}_p$ and $P_\phi$ is implicitly defined. Using $e^{\alpha_t P_\phi(\mathbf{G}_t,\lambda)}/Z_t$ in the property optimization task to maximize the target property is similarly done in Liu et al. [1] (the property function is in the exponent of the Boltzmann distribution).
> > > > > > > > >
> > > > > > > > > [1] Liu et al. GraphEBM: Molecular graph generation with energy-based models. ICLR Energy-Based Models Workshop, 2021.

---

### Official Review · Reviewer_gMda · 2022-10-24

**Confidence:** 3
**Correctness:** 2
**Technical Novelty And Significance:** 2
**Empirical Novelty And Significance:** 2
**Recommendation:** 5

**Clarity, Quality, Novelty And Reproducibility:**

Novelty is limited as it is a minor extension of the score-based generative model framework. Existing works for generating de novo drugs and satisfying desirable property constraints exist but rely on VAE rather than score-based models. Comparison with VAE-based approaches should be considered to support the reason why we need new solutions other than VAE-based solutions.


**Strength And Weaknesses:**

Strengths
The paper was well-written, it addresses a practical issue of AI-based approaches.

Weaknesses

Regarding the proposal of a controlled parameter for OOD sampling from a score-based model,  it is worth comparing the given approach to the VAE, where OOD sampling in VAE can be controlled by the deviation of the sampled latent variable to the predicted Gaussian distribution given by the encoders. I see a comparison to LIMO (Eckmann et al., 2022) but in the experiments with LIMO for novel drug generation, did you sample the latent variable using a constant factor deviating from the predicted standard deviation to control the novelty?


Drugs (Antimicrobial peptides) generation conditioned on a few desirable properties is not a new idea, please refer to Payes Das et al. (https://www.nature.com/articles/s41551-021-00689-x). The difference between the prior art is that it trains a conditional VAE to control the desirable property constraint while the (MOOD) framework is based on a score-based model and utilizes the gradient of a property prediction network to guide the sampling process to domains that are highly likely to satisfy the given constraints. I see a comparison to LIMO (Eckmann et al., 2022)  but the given method was not trained to generate molecules that have the constraint on desirable properties. This is not a fair comparison.


**Summary Of The Paper:**

Summary of the paper
The paper extends the score-based generative models by Song & Ermon (2019) to control the OOD level of generated molecules. To constrain the generation process such that it targets generating only OOD examples having desirable properties, they utilize the gradient of a property prediction network to guide the sampling process.

The authors demonstrated that the given approach outperformed the existing works on generating new drugs both in terms of novelty and having higher binding affinity confirmed by docking methods.



**Summary Of The Review:**

This is a minor extension of existing score-based generative models. Comparison with VAE models by Payes Das et al. (https://www.nature.com/articles/s41551-021-00689-x) on conditional generation and adjust the VAE to control the novelty in the latent space could be a potential simple baseline to compare to.

---

> ### Author Response · Authors · 2022-11-12
> **Initial Response**
>
> We sincerely thank you for your helpful comments. We appreciate your positive comments that the paper is well-written and aims to fill the gap between real-world drug discovery and AI-based approaches. We address all your concerns below.
>
> ---
>
> **Question 1:**
> OOD sampling in VAE can be controlled by the deviation of the sampled latent variable to the predicted Gaussian distribution given by the encoders. I see a comparison to LIMO but in the experiments with LIMO for novel drug generation, did you sample the latent variable using a constant factor deviating from the predicted standard deviation to control the novelty?
>
> **Answer 1:**
> We thank you for the interesting suggestion. Here we provide the results of LIMO with increased values of $\sigma$. With LIMO, we sampled latent variables from $\mathcal{N}(\bf{0}, \sigma^2\bf{I})$, where $\sigma=1$ in the original setting.
>
> **Table: Property optimization results with target protein parp1.**
> | Metric | LIMO ($\sigma=1$) | LIMO ($\sigma=1.01$) | LIMO ($\sigma=1.05$) | LIMO ($\sigma=1.1$) | MOOD (ours) |
> | --- | --- | --- | --- | --- | --- |
> | Novel hit ratio (%) | 0.455 | 0.189 | 0.233 | 0.167 | **7.017** |
> | Novel top 5% DS (kcal/mol) | -8.984 | -8.311 | -8.468 | -8.314 | **-10.865** |
>
> As shown in the table, increased variance in the VAE-based model does not help to generate novel, high-quality drugs. In fact, it degrades the performance since the ELBO objective minimizes KL divergence with the standard normal and the sampling distribution does not match it, and this scheme lacks theoretical ground unlike our proposed MOOD.
>
> ---
>
> **Question 2:**
> I see a comparison to LIMO but the given method was not trained to generate molecules that have the constraint on desirable properties. This is not a fair comparison.
>
> **Answer 2:**
> As described in Section C.4 of Appendix, we newly trained the property predictor of LIMO with respect to our target protein to make a fair comparison. We utilized the provided checkpoints for the base VAE model and the QED and SA predictors, and combined all the models strictly following LIMO’s procedure to generate molecules that have the desired properties. Therefore, we believe that the setting is fair.
>
> ---
>
> **Question 3:**
> Drug generation conditioned on a few desirable properties is not a new idea, please refer to Payes Das et al. [1]. Comparison with VAE-based approaches should be considered to support the reason why we need new solutions other than VAE-based solutions.
>
> **Answer 3:**
> In our paper, we compared our MOOD with three VAE-based methods: JTVAE (originally in Table 5 and 6; see Table 1 and 2 in the revised paper), HierVAE, and LIMO, and the first two are very popular models widely used as baselines. Moreover, compared to our MOOD and other baselines, HierVAE in our paper makes use of an active learning scheme which requires twice the expensive oracle calls as explained in Section C.4 of Appendix. Also, the mentioned paper [1] focuses on generating antimicrobial peptide sequences which is inapplicable to general molecule generation tasks. Therefore, we believe we provided a faithful comparison of MOOD with VAE-based approaches and proved that our MOOD significantly outperforms VAE-based methods in discovering novel hit molecules.
>
> ---
>
> **Question 4:**
> Novelty is limited as it is a minor extension of the score-based generative model framework.
>
> **Answer 4:**
> This is a critical misunderstanding about the contributions of our work. (1) We proposed a novel OOD-controlled reverse-time diffusion process that can control the amount of deviation from the training set, and (2) we proposed a novel conditional generation framework for molecule optimization that exploits the OOD-controlled process for extending the exploration space. To the best of our knowledge, we are the first to apply the concept of OOD in molecule generation to enhance novelty, and moreover our OOD-controlled diffusion process is universally applicable to the diffusion models of other domains. The framework is built on solid theoretical work with the basis of stochastic differential equations, and we further demonstrated its effectiveness experimentally. Note that the only score-based model for molecule generation is GDSS [2], which is an unconditional model that lacks the ability to optimize generated molecules with respect to target chemical properties, and further suffers from the explorability issue similar to previous generative models which we have shown in Figure 4 of our paper.
>
> ---
>
> **References**
>
> [1] Payel Das et al. Accelerated antimicrobial discovery via deep generative models and molecular dynamics simulations. Nature Biomedical Engineering 5.6 (2021): 613-623.
>
> [2] Jaehyeong Jo et al. Score-based Generative Modeling of Graphs via the System of Stochastic Differential Equations. ICML, 2022.

---

> > ### Comment · Reviewer_gMda · 2022-12-04
> > **Still keep my score**
> >
> > Dear authors,
> > Thank you for the answers to my questions. The new results reported in the table in your comments  made me more confused. The higher OOD-ness should be achieved by setting sigma for LIMO to be as large as possible, I don't understand why you have chosen sigma as 1.01, 1.05, and 1.1, why not varied from 0 to a larger value, say 100?
> > Also MOOD should have a hyperparameter to control the OOD-ness, you should show the results when that hyperparameter are varied. Since this is a multi-objective optimization problem, it requires a more careful analysis on results that show the relations between hyper-parameters and the multi-objective optimization metrics.
> > Regarding novelty, as I said people can control novelty in VAE via sigma, a more careful comparison with VAE-based methods is needed. Regarding conditional generation, it is not novel, please see the method from Payes Das et al. done with VAE.
> > I have also read the comments from both other reviews, since the novelty is limited and the results evaluation method is not convincing  I would like to keep my score as is.
> > regards

---

> > > ### Author Response · Authors · 2022-12-06
> > > **Response to Reviewer gMda**
> > >
> > > We thank you for replying to our response. We address your concerns below.
> > >
> > > ---
> > >
> > > **Question 1:**
> > > The higher OOD-ness should be achieved by setting sigma for LIMO to be as large as possible. I don't understand why you have chosen sigma as 1.01, 1.05, and 1.1, why not varied from 0 to a larger value, say 100? Regarding novelty, as I said people can control novelty in VAE via sigma, a more careful comparison with VAE-based methods is needed.
> > >
> > > **Answer 1:**
> > > As explained in the initial response, **increased variance in VAE-based models degrades the performance** since the training objective minimizes KL divergence with the standard normal in VAEs and the sampling distribution does not match it. As $\sigma$ more deviates from that in the training (i.e., $\sigma=1$), the performance degrades more, and larger $\sigma$ values like $100$ completely fail on the property optimization task because the distributional mismatch becomes larger. Therefore, **we cannot control novelty of the samples generated with VAE-based models by increasing the variance** since the scheme lacks theoretical ground unlike our proposed MOOD.
> > >
> > > ---
> > >
> > > **Question 2:**
> > > MOOD should have a hyperparameter to control the OOD-ness, you should show the results when that hyperparameter is varied.
> > >
> > > **Answer 2:**
> > > The effect of $\lambda$ on OOD-ness is shown in Figure 3 (for the ZINC250k dataset) and Table 4 of Appendix (for the QM9 dataset). For the effect of $\lambda$ on the property optimization task, we have shown the results with various values of $\lambda$ in the response to Reviewer xSWJ and also in Table 14 in the revised paper as below.
> > >
> > > **Table: Property optimization results with target protein parp1.**
> > > | $\lambda$ | Novelty (%) | Novel hit ratio (%) | Novel top 5% DS (kcal/mol) |
> > > | --- | --- | --- | --- |
> > > | 0.03 | 81.867 | 5.944 | -10.804 |
> > > | 0.04 | 84.180 | 7.017 | -10.865 |
> > > | 0.05 | 85.467 | 6.444 | -10.803 |
> > >
> > > As shown in the table, novelty increases as the value of $\lambda$ increases as in the novel molecule generation task. However, the results of $\lambda=0.03$ and $\lambda=0.05$ are worse than that of $\lambda=0.04$ in terms of novel hit ratio and novel top 5% DS. Intuitively, molecules that deviate from the training distribution naturally tend to be low-quality, and balancing the effect of the OOD control and the property gradient is important to produce novel and high-quality molecules. Empirically we found $\lambda=0.04$ performs reasonably well throughout the target proteins.
> > >
> > > ---
> > >
> > > **Question 3:**
> > > Regarding conditional generation, it is not novel, please see the method from Payes Das et al. done with VAE.
> > >
> > > **Answer 3:**
> > > Conditional generation is a very widely used strategy in property optimization tasks, and **we did not claim that the conditional generation itself is our contribution**.  **The reason we propose a conditional generation framework is to generate meaningful molecules from the out-of-distribution space, since without the conditional generation OOD generation will result in generating meaningless samples**. That is, without the conditional generation scheme, the generated OOD molecules may be chemically implausible, or may not meet the basic requirements of drugs. This is why we utilize the gradients from the property prediction network to steer the generation to the intersection of low-density regions and high-property regions, to generate novel but meaningful molecules.

---

> ### Author Response · Authors · 2022-11-25
> **Gentle reminder**
>
> We truly thank you for taking the time out of your busy schedule to participate in the review. We tried our best to answer your questions as below, and we hope your questions and concerns were addressed. Please review our response and let us know if you have more questions or comments.

---

### Official Review · Reviewer_3YM5 · 2022-10-25

**Confidence:** 3
**Correctness:** 2
**Technical Novelty And Significance:** 4
**Empirical Novelty And Significance:** 4
**Recommendation:** 5

**Clarity, Quality, Novelty And Reproducibility:**

## Clarity
This paper is clear enough and is easy to read.

## Quality
The method proposed is well aligned to the research objective of this paper, and is of high quality.

## Novelty
As far as I am aware of, the method disclosed in the paper is novel.

## Reproducibility
The source code is included in the supplementary material and is reproducible.

**Strength And Weaknesses:**

## Strengths
- The proposed method is a solid improvement over existing score-based graph generative models, whose effectiveness has been shown empirically.
- The evaluation metrics in the experiments are well designed to be aligned to the research objective, to find out-of-distribution molecules.
- Reproducibility is maintained as the supplementary material contains the code.

## Weaknesses
I am still not sure whether the generated molecules have to be out-of-distribution. A molecule similar to those in the dataset will be acceptable if its properties are sufficiently optimized; in other words, if we have two molecules that have similar properties but one of them is close to the training dataset while the other is far from the training dataset, then is the out-of-distribution molecule preferred? If not, then the novelty is not an end but a mean, and it is not natural to incorporate the novelty into the evaluation metric.

**Summary Of The Paper:**

The present paper is concerned about a generative model of molecules. The authors consider that the existing generative models are limited in that i) the generated molecules are similar to those in the training dataset, while what we want are very novel molecules, and ii) the target properties used in common benchmark tasks are not very helpful for drug discovery and existing methods are not likely to fully optimize more realistic multi-objective optimization tasks due to their limited exploration capabilities. Thus, the authors propose a molecular out-of-distribution diffusion (MOOD) framework, where the score-based model is guided by the property prediction network to generate out-of-distribution molecules with desirable properties.

In particular, the authors define the ODD-ness of a graph in Eq. (5) and use it to guide the generative process towards increased ODD-ness. The effectiveness of the ODD controller is showcased in Figure 2. In addition, the authors utilize property prediction networks to guide the generative process towards maximizing the properties. These two guidances are valanced by the magnitudes of the two related score functions.

In the experiments, the authors first confirm that their ODD controller can actually control the ODD-ness of the generated molecules. The authors then compare the proposed method with baseline methods in terms of novel hit ratio, which asks the method not only to optimize the objective function but also to be dissimilar to the molecules in the training dataset. The results show that the proposed method achieves the highest novel hit ratios. The authors also provide ablation studies.

**Summary Of The Review:**

I consider this paper is a border-line paper. While the method proposed is well designed to achieve the research objective, I am not fully convinced of the need of OOD-ness. It could be a mean to further optimize the objective function, but I am still not sure whether the OOD-ness itself is to be pursued or not. In addition, the property prediction or simulation-based property calculation could be inaccurate for OOD molecules, which de-motivate me to pursue OOD-ness. Therefore, to better assess the usefulness of this paper, I would like to request the authors to clarify i) whether the OOD-ness itself is pursued or it is just a mean to optimize the score, and ii) risks of pursuing OOD molecules and ways to avoid them. I would also appreciate it if the authors could point out my misunderstandings if any.

---

> ### Author Response · Authors · 2022-11-12
> **Initial Response**
>
> We sincerely thank you for your constructive comments. We appreciate your positive comments that both the technical and empirical novelty of our work are significant, the method and the evaluation metrics are well-designed, the paper is easy to understand, and our method is of high quality. We address your concerns below.
>
> ---
>
> **Question 1:**
> I am still not sure whether the generated molecules have to be out-of-distribution. If we have two molecules that have similar properties but one of them is close to the training dataset while the other is far from the training dataset, then is the out-of-distribution molecule preferred? If not, then the novelty is not an end but a mean, and it is not natural to incorporate the novelty into the evaluation metric.
>
> **Answer 1:**
> This is a critical misunderstanding as novelty is one of the most important and necessary considerations in *de novo* drug discovery. In the domain of drug discovery, it is a widely accepted concept that generating novel and diverse drug candidates are more desired than generating drug candidates with slight modifications from existing ones, and patents of novel drugs tend to be more valuable than patents of more derivative drugs [1, 2]. First, this is partly because there are a lot of requirements drug candidates need to satisfy other than binding affinity, and when designing drugs against a new disease, **novel core structures are often required to overcome major hurdles** that drug candidates encounter in later stages (molecules with similar structures show similar properties according to the structure-activity relationship (SAR)) [3, 4]. Second, the drug discovery pipeline includes the stage of lead generation and optimization, which slightly modifies discovered hit molecules for better target properties [5, 6, 7]. In this stage, **the derivatives of discovered hit molecules (the neighborhoods in the chemical space) are extensively investigated**. Therefore, *de novo* drug discovery, on which our work focuses, necessarily aims to broaden the horizons of known chemical space that can include novel hit or scaffold molecules that can ultimately be a solution to a new disease.
>
> ---
>
> **Question 2:**
> I would like to request the authors to clarify risks of pursuing OOD molecules and ways to avoid them.
>
> **Answer 2:**
> As we described in the Abstract and Introduction, the naïve OOD molecule generation can yield molecules that are chemically implausible, difficult to synthesize, and lacking desired properties, since the drug-like and high-quality chemical space is only a small portion of the vast chemical space. Therefore, we incorporated the conditional generation scheme in our MOOD framework to steer the OOD generation to the high-quality regions, and throughout our paper, we demonstrated that our method is indeed highly effective in generating novel, high-quality drug candidates.
>
> ---
>
> **References**
>
> [1] W. Patrick Walters and Mark Murcko. Assessing the impact of generative AI on medicinal chemistry. Nature biotechnology 38.2 (2020): 143-145.
>
> [2] Joshua Krieger et al. Missing novelty in drug development. The Review of Financial Studies 35.2 (2022): 636-679.
>
> [3] Thorsten Meinl et al. Maximum-score diversity selection for early drug discovery. Journal of chemical information and modeling 51.2 (2011): 237-247.
>
> [4] Hongyu Zhao and Irini Akritopoulou-Zanze. When analoging is not enough: scaffold discovery in medicinal chemistry. Expert Opinion on Drug Discovery 5.2 (2010): 123-134.
>
> [5] Daria Mochly-Rosen and Kevin Grimes, A practical guide to drug development in academia. Springer Science & Business Media, 2014.
>
> [6] Hongyu Zhao. Scaffold selection and scaffold hopping in lead generation: a medicinal chemistry perspective. Drug discovery today 12.3-4 (2007): 149-155.
>
> [7] Leïla Abrous, et al. Design and Synthesis of Novel Scaffolds for Drug Discovery: Hybrids of β-d-Glucose with 1, 2, 3, 4-Tetrahydrobenzo [e][1, 4] diazepin-5-one, the Corresponding 1-Oxazepine, and 2-and 4-Pyridyldiazepines. Organic Letters 3.7 (2001): 1089-1092.

---

> > ### Comment · Reviewer_3YM5 · 2022-11-19
> > **Re: Initial response**
> >
> > I am still not very convinced of pursuing OOD-ness first. While the authors argue that,
> > > when designing drugs against a new disease, novel core structures are often required to overcome major hurdles that drug candidates encounter in later stages (molecules with similar structures show similar properties according to the structure-activity relationship (SAR)) [3, 4].
> > I am not an expert on drug discovery and therefore I cannot judge whether this is true or not. For example, let us consider a novel molecule that satisfies the requirements but is similar to known molecules. If the known molecule turns out to be approved by the authority as a drug, then the authors' statement "molecules with similar structures show similar properties" guarantee that such a novel molecule is likely to be approved. I acknowledge that OOD-ness can be one heuristics to discover novel and high-value molecules, but it is not a silver bullet. At least, I would appreciate it if the authors clarify the benefits and risks of pursuing OOD-ness, as compared to just optimizing the objective functions.

---

> > > ### Author Response · Authors · 2022-11-21
> > > **Response to Reviewer 3YM5**
> > >
> > > We sincerely thank you for replying to our response. We address your concerns below.
> > >
> > > **Question:**
> > > If the known molecule turns out to be approved by the authority as a drug, then the authors' statement "molecules with similar structures show similar properties" guarantee that such a novel molecule is likely to be approved.
> > >
> > > **Answer:**
> > > This is a **critical misunderstanding on the drug discovery pipeline**. In drug discovery, our goal is to discover a drug **against a new disease or a with new effect**. **There is little need to re-discover a drug with the same chemical properties with existing molecules**. Otherwise, we wouldn’t need to solve the problem. Most known chemical databases are composed of active molecules against a target protein (i.e., those show high affinity; this does not mean they are approved drugs against a target disease) or approved drugs against other diseases. By pursuing OOD-ness, we set an implicit assumption that there are no drugs with acceptable side-effects in the training set, and thus to discover a drug with novel bioactivities or mitigated side-effects, we need to consider structural novelty. This is the main reason why chemists try to discover novel and diverse drug candidates in the early stages of drug discovery.

---

### Decision · Program_Chairs · 2023-01-20

**Decision:**

Reject

**Justification For Why Not Higher Score:**

There are too many open questions ranging from the over-all usefulness of the OOD approach for the generation of molecules to concerns about the novelty and significance of his work to formal questions about the normalization of probability distributions. Many of these problems could not be resolved in a convincing way during the discussion phase.

**Justification For Why Not Lower Score:**

N/A

**Metareview: Summary, Strengths And Weaknesses:**

There are two borderline (negative) reviews, in which certain strengths and weaknesses are mentioned.
On the positive side, most reviews seem to agree that  this paper is generally well-written, and that it addresses an important problem.
Negative aspects mentioned in these two "borderline" reviews include severe questions about the motivation for an OOD approach in the context of molecule generation, the relation to VAE-based OOD approaches, and the experimental evaluation method, that was considered as  not fully convincing. Although some of these concerns could be addressed to some degree in the rebuttal, none of these reviewers were finally too impressed about the rebuttal and the paper in general, and they wanted to keep their (slightly) negative scores.

There was a third reviewer, who wrote a clearly negative review (which even became more negative during the discussions). I do understand and mostly agree with the main points of criticism raised by this reviewer, such as the question about the proper normalization of the  joint distribution of G and lambda, which was never answered in a clear and transparent way by the authors. I find this a but puzzling, since the authors claim at the same time that this normalization was obvious. Although I think that the final score of 1 assigned by this reviewer is too low, I fully agree that there is an important open question, which is problematic, since it directly addresses the underlying methodological concept.

A fourth reviewer initially had a much more positive impression of this paper, particularly about the idea that the  OOD-ness can be adjusted by a parameter. At the end of the discussion period, however, also this reviewer saw several flaws with this submission that have been mentioned in the other reviews, and did not want to champion this paper for acceptance.

In summary, none of the initially rather neutral reviewers could be convinced to assign a positive score, and in the end also the most positive reviewer shared several points of criticism raised by the other reviewers. For me, the negative arguments finally dominated, and therefore I recommend rejection.